M. Bocher A. Fournier N. Coltice

# Ensemble Kalman filter for the reconstruction of the Earth's mantle circulation

Marie Bocher[1,2], Alexandre Fournier[3], and Nicolas Coltice[2]

[1]Computational Seismology, Institute of Geophysics, Department of Earth Sciences, ETH Zürich, Sonneggstrasse 5, CH-8092 Zürich, Switzerland.
[2]Laboratoire de Géologie de Lyon, Université Claude Bernard Lyon 1, Ecole Normale Supérieure de Lyon, France.
[3]Institut de Physique du Globe de Paris, Sorbonne Paris Cité, Université Paris Diderot,CNRS (UMR 7154), Paris, France

*Correspondence to:* M. Bocher (mariebocher@gmail.com)

**Abstract.** Recent advances in mantle convection modelling led to the release of a new generation of convection codes, able to generate self-consistently plate-like tectonics at their surface. Those models physically link mantle dynamics to surface tectonics. Combined with plate tectonic reconstructions, they have the potential to produce a new generation of mantle circulation models that use data assimilation methods and where uncertainties on plate tectonic reconstructions are taken into account. We recently provided a proof of this concept by applying a suboptimal Kalman Filter to the reconstruction of mantle circulation (Bocher et al., 2016). Here, we propose to go one step further and apply the ensemble Kalman filter (EnKF) to this problem. The EnKF is a sequential Monte Carlo method particularly adapted to solve high dimensional data assimilation problems with nonlinear dynamics. We tested the EnKF using synthetic observations consisting of surface velocity and heat flow measurements, on a 2D-spherical annulus model and compared it with the method developed previously. The EnKF performs on average better and is more stable than the former method. Less than 300 ensemble members are sufficient to reconstruct an evolution. We use covariance adaptive inflation and localization to correct for sampling errors. We show that the EnKF results are robust over a wide range of covariance localization parameters. The reconstruction is associated with an estimation of the error, and provides valuable information on where the reconstruction is to be trusted or not.

## 1 Introduction

Mantle circulation models are estimates of mantle flow history. They combine two sources of information: observations on the dynamics or 3D structure of the Earth's mantle and a numerical model of mantle convection. In their effort to reconcile both observations and our physical understanding of mantle dynamics, they serve a wide variety of purposes and disciplines. Hager and O'Connell (1979) originally built instantaneous mantle circulation models to understand the effect of plates on large-scale mantle flow. Since then, they have been used, among other applications, to understand the dynamics and evolution of the deep earth mantle structures (Bunge et al., 1998; McNamara and Zhong, 2005; Bower et al., 2013; Davies et al., 2012), to study the evolution of mantle plumes and their relationship to hotspots (Hassan et al., 2016), to infer changes in the Earth's rotation axis (Steinberger and O'Connell, 1997), sea-level (Moucha et al., 2008) or dynamic topography (Flament et al., 2013).

The geodynamics community has developed three alternative approaches to the problem of the reconstruction of mantle circulation. The first approach, backward advection, consists in starting at present by estimating the current density field of the mantle from seismic tomography models (see Conrad and Gurnis, 2003, for a description of this method). This density field is then advected backward in time with plate tectonic reconstructions as imposed boundary condition (Steinberger and O'Connell, 1997). This method has a limited numerical cost and exploits the two most instructive constraints on mantle circulation: plate tectonic reconstructions and seismic tomography. However, this technique neglects thermal diffusion, so it is not able to reconstruct past thermal structures that have completely diffused before present and it is limited to times and regions for which the effect of diffusion is thought to be small. This limits reconstructions to the last 50 to 75 Ma (Conrad and Gurnis, 2003) or even to shorter periods if we consider the uncertainties on tomographic models (Bello et al., 2014). The second approach, the semi-empirical sequential method, estimates mantle circulation by integrating plate tectonic reconstructions chronologically into a mantle convection model. Plate tectonic reconstructions are either introduced as velocity boundary conditions, as first described by Bunge et al. (1998), or with a more sophisticated method, by blending a convection solution with thermal and kinematic models of plates and slabs (Bower et al., 2015). This approach allows the use of models of convection with chemical heterogeneities (McNamara and Zhong, 2005). Also, it is not anymore the reconstruction method that limits the timespan of the reconstruction, but the availability of plate tectonic reconstructions. This led to mantle circulation models integrating up to 450 Ma of plate reconstruction history (Zhang et al., 2010). However, this method considers plate tectonic reconstructions as perfect estimates of surface tectonics: uncertainties affecting the reconstructions are not taken into account although they are substantial, especially as reconstructions go further in the past (for example, there is almost no information on the state of the ocean floor before 140 Ma, see e.g. Torsvik et al., 2010). This method also requires the choice of an arbitrary initial temperature field to compute the evolution. The third approach uses data assimilation methods to solve the mantle circulation problem. Data assimilation methods are inverse methods dealing with the specific problem of estimating the evolution of a dynamical system from asynchronous data and a physical model (Evensen, 2009a). The full inverse problem for mantle circulation, as stated by Bunge et al. (2003), would take into account model errors, numerical approximations, errors on plate reconstructions and on the estimation of the current tomography-derived temperature field to provide the best fit given all sources of information. However, solving the full inverse problem of mantle circulation is still a great challenge given the nonlinearities in mantle convection dynamics and the computational power required to compute a realistic forward mantle convection evolution alone (Stadler et al., 2010; Burstedde et al., 2013). So far, variational data assimilation dominates over other methods to estimate mantle circulation (Bunge et al., 2003; Horbach et al., 2014; Ghelichkhan and Bunge, 2016). To simplify the problem, they minimize the misfit between the final temperature field of the mantle circulation model and the one deduced from seismic tomography. These mantle circulation models impose plate tectonic reconstructions as boundary conditions, as in the first two approaches.

Here, we take a different view on data assimilation methods for mantle circulation models by focusing on how to take into account the uncertainties in plate tectonic reconstructions. For almost a decade, 3-D spherical mantle convection models have shown the capability to self-consistently produce plate-like tectonics at their surface (Walzer and Hendel, 2008; Van Heck and Tackley, 2008; Yoshida, 2008; Foley and Becker, 2009). These models physically link surface tectonics comparable to that of

the Earth to mantle convection processes (Coltice et al., 2012; Rolf et al., 2014; Mallard et al., 2016). In Bocher et al. (2016), we took advantage of this link to build a sequential data assimilation algorithm able to integrate plate reconstructions into a mantle convection code while taking into account the uncertainties on those plate tectonic reconstructions. This technique assimilates a time series of surface observations chronologically, by repeating two stages, analysis and forecast, until all observations are taken into account. Whenever an observation is available, the analysis evaluates the most likely state of the mantle at this time, considering a prior guess (supplied by the forecast) and the new observations at hand. For this evaluation, we used the classical best linear unbiased estimate (Talagrand, 1997). Then, the forward model of mantle convection forecasts the evolution of the mantle until the next observation time. We tested this algorithm on synthetic experiments. It proved to be efficient in recovering mantle circulation given constraints on the amplitude of errors affecting observations and the timespan between observations.

Here we extend this work by applying a more advanced sequential data assimilation method, the ensemble Kalman filter (EnKF, originally described in Evensen, 1994, and in its corrected version in Burgers et al., 1998). This method is particularly suited for high dimensional nonlinear dynamical models (Evensen, 2009b). Instead of estimating the most likely state of the mantle, the ensemble Kalman filter provides at each time an approximation of the probability density function of the state of the system in the form of a finite ensemble of states. During the forecast stage, each member of the ensemble evolves independently. For the analysis, we use the second order statistics of the ensemble to correct each ensemble member with the new observations at hand. We evaluate this method with synthetic experiments in 2D-spherical annulus geometry (Hernlund and Tackley, 2008) and compare it to the algorithm developed by Bocher et al. (2016). The EnKF provides more accurate estimations than the former method, and is even able to reconstruct evolutions that the former method could not. Moreover, the EnKF also estimates locally the error on the reconstruction. Both covariance inflation and localization eliminate spurious correlations arising from the finite size of the ensemble that is used to compute them.

This paper is organized as follows. In section 2, we present our simplifications on the general mantle circulation reconstruction problem and the correspondence with the notation in the EnKF algorithm. Then, in section 3, we detail the EnKF method and justify the variants chosen for the application to mantle circulation. Section 4 presents the results obtained on synthetic experiments and compares them to results obtained by the method described in Bocher et al. (2016). Section 5 is a discussion on the choice of the method and the challenges involved in the application of such a method to a realistic setting.

## 2 Presentation of the problem

We aim at reconstructing mantle circulation for the last hundreds of millions of years by combining a mantle convection model with plate tectonic reconstructions, using an ensemble Kalman filter. To study the behavior of the ensemble Kalman filter on such problem, we consider a simplified mantle convection model. This section describes the model used to compute a mantle evolution, the data set assimilated in this evolution, and finally the backbone of ensemble Kalman filtering.

## 2.1 Mantle convection model

At the timescales and lengthscales we are interested in ($\geq 10$ kyr, $\geq 1000$ km), the mantle can be modelled as a continuous viscous medium. To compute mantle circulation, we solve the equations of conservation of mass (Eq. (1) below), momentum (Eq. (2) below) and energy (Eq. (8) below) for an isochemical mantle under the Boussinesq approximation. The system of equations is non-dimensionalized to the thermal diffusion time scale (see Ricard, 2015). Given the high Prandtl number of the mantle (of the order of $10^{24}$), inertia is neglected. With these assumptions, the equations of conservation of mass and momentum become diagnostic equations of the form

$$\nabla \cdot \boldsymbol{u} = 0, \tag{1}$$

$$\nabla \cdot \boldsymbol{\sigma} - \nabla p + \mathrm{Ra}_T T \boldsymbol{e}_r = 0, \tag{2}$$

where $\boldsymbol{\sigma}$, $\boldsymbol{u}$, $p$, and $T$ are the non-dimensional deviatoric stress, velocity, dynamic pressure, and temperature, respectively. $\mathrm{Ra}_T$ is the Rayleigh number based on the temperature difference between the top and bottom boundaries of the domain, defined as

$$\mathrm{Ra}_T = \frac{\rho_0 g_0 \alpha_0 \Delta T a^3}{\mu_0 \kappa_0} \tag{3}$$

with $\rho_0$ the density for $T = 0$, $g_0$ the gravitational acceleration, $\alpha_0$ the thermal expansivity, $\Delta T$ the temperature drop, $a$ the depth of the layer, $\kappa_0$ the thermal diffusivity, $\mu_0$ the dynamic viscosity of the system. The vertical velocities and shear-stress at the surface and the base of the model are set to zero.

The deformation response of mantle material to stress is implemented as a linear relationship linking the strain rate tensor $\dot{\boldsymbol{\epsilon}}$ to the deviatoric stress tensor $\boldsymbol{\sigma}$ as

$$\boldsymbol{\sigma} = 2\mu_{\mathrm{eff}}\dot{\boldsymbol{\epsilon}} = \mu_{\mathrm{eff}}\left(\nabla \boldsymbol{u} + (\nabla \boldsymbol{u})^T\right). \tag{4}$$

The choice of the effective viscosity $\mu_{\mathrm{eff}}$ is crucial for the development of plate-like tectonics at the surface of the convective system. We choose for $\mu_{\mathrm{eff}}$ a composite rheology with a viscous Newtonian component $\mu_n$ and a pseudo-plastic component, implemented with an equivalent "pseudo-plastic viscosity" $\mu_y$, such that

$$\mu_{\mathrm{eff}} = \min(\mu_n, \mu_y). \tag{5}$$

The Newtonian viscosity $\mu_n$ follows an Arrhenius law

$$\mu_n = \mu_0 \exp\left(\frac{E_A}{T + T_1}\right) \tag{6}$$

with $\mu_0 = \exp\left(-\frac{E_A}{2T_1}\right)$, $T_1$ the temperature at which the nondimensional $\mu_n = 1$, and $E_A$ the nondimensional activation energy. This law reflects the thermal activation of crystal deformation, and creates a highly viscous upper boundary layer (the lithosphere), while the rest of the mantle is less viscous. We also implement the decrease of viscosity in the asthenosphere (the layer below the lithosphere) by reducing by a factor of 10 the viscosity $\mu_n$ when the temperature is above a solidus equation

$T_s = T_{s_0} + \nabla_r T_s (r_a - r)$ with $r_a$ the surface value of $r$. The presence of a weak asthenosphere tends to favor plate-like behavior (Tackley, 2000; Richards et al., 2001), and is compatible with laboratory and observational data (King, 2016).

The pseudo-plastic part $\mu_y$ is defined by

$$\mu_y = \frac{\sigma_{yield}}{2\dot{\epsilon}_{\mathrm{II}}}, \tag{7}$$

where $\dot{\epsilon}_{\mathrm{II}}$ is the second invariant of the strain rate tensor and $\sigma_{yield} = \sigma_Y + (r_a - r)\nabla_r \sigma_Y$, with $\sigma_Y$ and $\nabla_r \sigma_Y$ the yield stress at the surface and the depth-dependence of the yield stress, respectively.

This composite rheology allows the development of strong plates delimited by narrow weak zones (i.e. plate boundaries), and is currently the best way to generate self-consistently plate-like tectonics at the surface of global mantle convection models (Coltice et al., 2017).

The energy conservation equation is the only prognostic equation of the system

$$\frac{\mathrm{D}T}{\mathrm{D}t} = \nabla^2 T + \frac{\mathrm{Ra}_H}{\mathrm{Ra}_T}, \tag{8}$$

with $\mathrm{Ra}_H$ the Rayleigh number based on internal heating

$$\mathrm{Ra_H} = \frac{\rho_0^2 g_0 \alpha_0 H a^5}{\mu_0 k_0 \kappa_0} \tag{9}$$

with $H$ the dimensional heating rate and $k_0$ the thermal conductivity. We set isothermal top and bottom boundaries with temperatures $T_a$ and $T_b$, respectively. The models presented here have 10% basal heating and 90% internal heating.

These equations are solved using the finite volume, multigrid parallel code STAGYY (Tackley et al., 1993), on a spherical annulus staggered grid. This geometry provides results closer to the spherical geometry than cylindrical geometry (Hernlund and Tackley, 2008). In the following, the longitudinal coordinate of a point is $\phi_l$, with $l \in \{1, 2, ..., L\}$ and its radial coordinate is $r_m$ with $m \in \{1, 2, ..., M\}$, $r$ varying from $r_b$ to $r_a$.

Note that this paper focuses on the methodology of ensemble data assimilation for a convecting system similar to that of the Earth's mantle. Hence, we choose a rather simple model that can reproduce plate-like tectonics at the surface. We rely on simplifications such as 2D geometry, incompressible and isochemical mantle and a rheology which does not take into account the history of the material. Although some of the complexities we ignore may play a fundamental role in the reconstruction of the Earth's mantle evolution, we choose to focus in this manuscript on the data assimilation methodology. Moreover, we choose to keep the same parameters as the test case of Bocher et al. (2016) in order to enable direct comparison between the methods. Table 1 lists the chosen parameter values.

To ease the comparison with Earth's mantle convection, we rescale the non-dimensional time in the evolution, $t$, by the transit time of the convective system. By definition, the transit time of the Earth's mantle is $t_t^E = a^E / v_{rms}^E$, with $a^E$ the thickness of the mantle and $v_{rms}^E$ the root mean square of surface velocities of the Earth, as estimated by plate tectonic reconstructions (Seton et al., 2012). We compute the same quantity for the model $t_t^m = a / v_{rms}^m$. The scaled time $t^s$ is then $t^s = t \frac{t_t^E}{t_t^m}$.

The dynamics of the convective system we just described depends on the two dimensionless numbers $\mathrm{Ra}_T$ and $\mathrm{Ra_H}$. In our model, $\mathrm{Ra}_T = 10^6$ and $\mathrm{Ra_H} = 2.05 \, 10^7$. These values are one to two orders of magnitude lower than the current Earth

estimates, but high enough to ensure chaotic convection with thermal turbulence (Stewart and Turcotte, 1989; Travis and Olson, 1994). In this regime, the top and bottom boundary layers develop instabilities that can trigger transient descending and ascending currents, respectively. This leads to a highly time-dependent flow, and the exponential growth of perturbations of the initial state of the system, as studied by Bello et al. (2014), in a series of twin experiments in 3D spherical geometry.

We computed the Lyapunov time corresponding to the time over which initial perturbations grow exponentially by a factor of $e$, and found for our models a lyapunov time of 140 Myr, similar to the times Bello et al. (2014) estimated for their most Earth-like model.

## 2.2   Observations of mantle circulation

The state of the Earth's surface is the time integrated expression of mantle circulation. At a global scale, the main source of
information for the last 100 Myr is the database of the localization and identification of magnetic anomalies on the seafloor, translated into maps of seafloor ages (Müller et al., 2008; Seton et al., 2014). This information is complemented with regional geological studies giving constraints on the timing and geometry of tectonic events as well as a synthesis of paleontological, structural geology, stratigraphical, magnetic anomalies, gravity data and seismic studies. In addition, paleomagnetic data provide constraints on the paleolatitude of continental blocks (Besse and Courtillot, 2002).

Plate tectonic reconstructions use the geometric theory of plate tectonics to integrate all these observations. The result is a time series of maps of seafloor ages, plate layout and kinematics. The continuously closed plate algorithm (Gurnis et al., 2012) produces plate tectonic reconstruction maps continuous in space and time (Seton et al., 2012; Müller et al., 2016).

Although we are aware that these plate tectonic reconstruction maps are in themselves models and not direct observations, we propose to develop an assimilation method that use them as data to assimilate in our mantle convection model. This solution
is generally chosen in mantle circulation reconstructions (Bunge et al., 2002; Zhang et al., 2010; Bower et al., 2015), because it provides continuous surface boundary conditions in space and time for the period of reconstruction. One advantage of the technique we develop is that it is possible to consider errors on the data that are assimilated, another is that the reconstructions do not need to be known at all times and at all points on the surface. Hence it is possible, in principle, to design a data assimilation scheme using direct observations. However, this would require further developments both on the database design
and on the data assimilation algorithm. Sequential data assimilation methods for mantle circulation are still in their infancy, so we opt for a simpler structure of the data to be assimilated: a time series of maps of surface velocity and seafloor age, as given by plate tectonic reconstructions.

In this study, we limit ourselves to the test of data assimilation in synthetic experiments. In the model described in Sect. 2.1, the absence of small scale convection at the base of the boundary layer makes the surface heat flux an excellent proxy for
the age of the seafloor (Coltice et al., 2012). Consequently, we consider surface heat flux and surface velocity as the data to assimilate.

To our knowledge, the amplitude of the uncertainty on global plate tectonic reconstructions has not yet been assessed. For the synthetic tests we perform in Sect. 4, we choose an arbitrary value of 10% of the root mean square value of heat flux and surface velocity, respectively. We further discuss this choice in Sect. 5.

## 2.3 Ensemble Kalman filtering framework: notations

Our aim is to assimilate a time series of observations (surface velocities and heat fluxes) into a mantle convection model to estimate the evolution of the state of the mantle. We introduce here the general formulation of ensemble Kalman filtering and link them to our problem. We use the notation system recommended by Ide et al. (1997).

The time series of data is defined as a set of column vectors $\{\boldsymbol{y}_1^o, \boldsymbol{y}_2^o, ..., \boldsymbol{y}_K^o\}$, where the subscripts $\{1, 2, ..., K\}$ refer to the times at which observations are available. As seen in the previous section, the data used for our experiments are surface velocity and surface heat flux. The data vector at time $k$ is thus defined as

$$\boldsymbol{y}_k^o = \left[q_k^o(\phi_1), q_k^o(\phi_2), ..., q_k^o(\phi_L), u_{\phi k}^o(\phi_1), u_{\phi k}^o(\phi_2), ..., u_{\phi k}^o(\phi_L)\right]^T, \tag{10}$$

where $q_k^o(\phi_l)$ and $u_{\phi k}^o(\phi_l)$ are the observed values of surface heat flux and surface horizontal velocity at the $k$-th timestep and

longitude $\phi_l$, and $(\cdot)^T$ means transpose. We model errors on observations by a random vector of zero mean and covariance matrix $\mathbf{R}_k$ (we suppose unbiased observations). Although $\mathbf{R}_k$ is a diagonal matrix of constant value and size in our experiments, it is not generally the case. Correlations between errors on observations could be specified in $\mathbf{R}_k$.

The evolution of the state of the system is estimated sequentially during the period where observations are available. At each timestep $k \in \{1, 2, ..., K\}$, we define two state vectors: the a priori state, or forecast state $\boldsymbol{x}_k^f$ and the analysis state $\boldsymbol{x}_k^a$, which

is the state corrected after having assimilated the observations $\boldsymbol{y}_k^o$. The system of equations developed in Sect. 2.1 shows that we can compute velocity, viscosity and pressure values at each grid point from the sole knowledge of the temperature field: the temperature field describes completely the state of the system. However, the relation between surface velocities and the temperature field is nonlinear. We choose to include in the state the whole temperature field, but also add the surface velocities, to form an augmented state vector, following the suggestion of Evensen (2003), Sect. 4.5. This formulation establishes a linear

relationship between the state and data (see last paragraph of this section), which simplifies the computations thereafter. The state of the mantle at a timestep $k \in [1, K]$ is defined as

$$\boldsymbol{x}_k = \left[T_k(\phi_1, r_1), T_k(\phi_1, r_2), ..., T_k(\phi_L, r_M), u_{\phi k}(\phi_1), u_{\phi k}(\phi_2), ..., u_{\phi k}(\phi_L)\right]^T, \tag{11}$$

where $T_k(\phi_l, r_m)$ and $u_{\phi k}(\phi_l)$ are the values of temperature at the $k$th timestep, longitude $\phi_l$ and radius $r_m$ and surface horizontal velocity at the $k$th timestep and longitude $\phi_l$.

The forecast and analyzed states are uncertain as well. Their uncertainties are represented by two random vectors of zero expected value and covariance matrices $\mathbf{P}_k^f$ and $\mathbf{P}_k^a$, respectively. We compute explicitly these covariance matrices only for the initialization step (see Sect. 3.1). Otherwise, the uncertainty on the forecast and analyzed states is represented by two ensembles of $N$ states $\{\boldsymbol{x}_{kn}^f\}_{n \in [1,N]}$ and $\{\boldsymbol{x}_{kn}^a\}_{n \in [1,N]}$, such that their average equals $\boldsymbol{x}_k^f$ and $\boldsymbol{x}_k^a$, respectively, and their respective sample covariance matrices approximate $\mathbf{P}_k^f$ and $\mathbf{P}_k^a$. The ensemble of states $\{\boldsymbol{x}_{kn}^f\}_{n \in [1,N]}$ and $\{\boldsymbol{x}_{kn}^a\}_{n \in [1,N]}$ are stored in the matrices

$\mathbf{X}_k^f$ and $\mathbf{X}_k^a$, where the $n$th column is the state of the $n$th ensemble member $\boldsymbol{x}_{kn}^f$ and $\boldsymbol{x}_{kn}^a$, respectively.

Finally, we introduce the observation operator, which maps a given state vector $\boldsymbol{x}_{kn}^e$ ($e$ being $f$ or $a$) to the corresponding data $\boldsymbol{y}_{kn}^e$. The surface heat flux is approximated by a first order discretization of Fourier's law. The observation operator is then

linear, with its velocity part being simply the identity, and can be represented by the matrix $\mathbf{H}$ such that

$$\forall k \in \{1, 2, ..., K\}, \forall n \in \{1, 2, ..., N\}, \quad \boldsymbol{y}_{kn}^e = \mathbf{H}\boldsymbol{x}_{kn}^e. \tag{12}$$

Table 2 summarizes the dimensions of the vectors and matrices for our problem.

## 3 Ensemble Kalman filter with localization and inflation

The ensemble Kalman filter (Evensen, 1994; Burgers et al., 1998) is a sequential data assimilation algorithm using the same equations as the Kalman Filter for the analysis step, but Monte Carlo methods to forecast the error statistics on the state. We explain here how we adapt the ensemble Kalman filter to our problem and justify the choice of the starting ensemble.

To implement the EnKF, we used the software environment Parallel Data Assimilation Framework (PDAF, Nerger et al., 2005; Nerger and Hiller, 2013).

### 3.1 Initialization: first analysis and generation of the starting ensemble

We compute the second order statistics of the background state from a series of $400$ decorrelated snapshots of convection simulations by following the procedure detailed in Bocher et al. (2016), Sect. 4.1. The model setup is spherically symmetric, so the expected value and covariance of the background temperatures and surface velocities must satisfy

$$\forall(\phi, r), \ \langle T(\phi, r) \rangle = \langle T(0, r) \rangle, \tag{13}$$

$$\forall(\phi_1, \phi_2, r_1, r_2), \ \mathrm{Cov}(T(\phi_1, r_1), T(\phi_2, r_2)) = \mathrm{Cov}(T(0, r_1), T(\phi_1 - \phi_2, r_2)), \tag{14}$$

$$= \mathrm{Cov}(T(0, r_1), T(\phi_2 - \phi_1, r_2)), \tag{15}$$

where $\langle \cdot \rangle$ stands for the expectation operator and $\mathrm{Cov}(\cdot, \cdot)$ stands for covariance operator. Likewise, we have

$$\forall \phi, \ \langle u_\phi(\phi) \rangle = \langle u_\phi(0) \rangle, \tag{16}$$

$$\forall(\phi_1, \phi_2), \ \mathrm{Cov}(u_\phi(\phi_1), u_\phi(\phi_2)) = \mathrm{Cov}(u_\phi(0), u_\phi(\phi_1 - \phi_2)), \tag{17}$$

$$\forall(\phi_1, \phi_2, r_1), \ \mathrm{Cov}(T(\phi_1, r_1), u_\phi(\phi_2)) = \mathrm{Cov}(T(0, r_1), u_\phi(\phi_2 - \phi_1)), \tag{18}$$

$$= -\mathrm{Cov}(T(0, r_1), u_\phi(\phi_1 - \phi_2)). \tag{19}$$

We use these symmetries to compute

$$\langle T(0, r_m) \rangle, \ \text{with } m \in \{1, ..., M\} \tag{20}$$

$$\mathrm{Cov}(T(r_m), T(\phi_{l'}, r_{m'})), \ \text{with } m \in \{1, ..., M\}, l' \in \{1, ..., L/2\}, \text{ and } m' \in \{1, ..., M\} \tag{21}$$

$$\langle u_\phi(0) \rangle, \tag{22}$$

$$\mathrm{Cov}(u_\phi(0), u_\phi(\phi_l)), \ \text{with } l \in \{1, ..., L/2\} \tag{23}$$

$$\mathrm{Cov}(u_\phi(0), T(\phi_l, r_m)), \ \text{with } l \in \{1, ..., L/2\}, \text{ and } m \in \{1, ..., M\}, \tag{24}$$

and build with these values the first forecast state of expected value $\boldsymbol{x}_1^f$ and associated covariance matrix $\mathbf{P}_1^f$. For the model used in this study (see Table 1), the covariance matrix $\mathbf{P}_1^f$ has $(LM+L)^2 = 18,816^2 = 354,041,856$ components. By using the symmetries in the system, we are able to reduce the number of independant components in the covariance matrix to $L/2(M+1)^2 = 3,557,400$. $\mathbf{P}_1^f$ is eigendecomposed and rank reduced into $\mathbf{P}_{1r}^f = \mathbf{V}\boldsymbol{\Lambda}\mathbf{V}^T$, with $\boldsymbol{\Lambda}$ a diagonal matrix containing the $n_r = 1928$ largest eigenvalues of $\mathbf{P}_1^f$ (which accounts for 99.98% of its cumulative variance) and $\mathbf{V}$ the corresponding matrix of eigenvectors.

We assimilate the first set of observations $\boldsymbol{y}_1^o$ using the classical Best Linear Unbiased Estimator equations (see Ghil and Malanotte-Rizzoli (1991) for example). When the forecast covariance matrix is eigendecomposed and rank reduced, these equations can take the form

$$\boldsymbol{x}_1^a = \boldsymbol{x}_1^f + \mathbf{V}\mathbf{A}\mathbf{V}^T\mathbf{H}^T\mathbf{R}^{-1}(\boldsymbol{y}_1^o - \mathbf{H}\boldsymbol{x}_1^f), \tag{25}$$

$$\mathbf{P}_1^a = \mathbf{V}\mathbf{A}\mathbf{V}^T, \tag{26}$$

with

$$\mathbf{A} = \left[\boldsymbol{\Lambda}^{-1} + \mathbf{V}^T\mathbf{H}^T\mathbf{R}^{-1}\mathbf{H}\mathbf{V}\right]^{-1}. \tag{27}$$

After the first analysis, we generate an ensemble of $N$ initial states from the first analyzed state average $\boldsymbol{x}_1^a$ and associated covariance matrix $\mathbf{P}_1^a$. To do so, we follow the second order exact sampling method (Hoteit, 2001; Pham, 2001). First, A is eigendecomposed

$$\mathbf{A} = \mathbf{V}^a\boldsymbol{\Lambda}^a\mathbf{V}^{aT}. \tag{28}$$

The ensemble members are then computed following

$$\mathbf{X}_1^a = \begin{pmatrix} | & & | \\ \boldsymbol{x}_{11}^a & \dots & \boldsymbol{x}_{1N}^a \\ | & & | \end{pmatrix} = \begin{pmatrix} | & & | \\ \boldsymbol{x}_1^a & \dots & \boldsymbol{x}_1^a \\ | & & | \end{pmatrix} + \sqrt{N-1}\mathbf{V}\mathbf{V}^a\boldsymbol{\Lambda}^{a1/2}\begin{pmatrix} \boldsymbol{\Omega}_{N\times(N-1)}^T \\ \mathbf{0}_{(n_r-N)\times N} \end{pmatrix}, \tag{29}$$

where $\boldsymbol{\Omega}_{N\times(N-1)}$ is a random matrix whose columns are vectors forming an orthonormal basis and each of them is orthogonal to $\mathbf{1}_N$, the column vector of dimension $N$ full of 1, $\mathbf{1}_N = [1,...,1]^T$. $\mathbf{0}_{(n_r-N)\times N}$ is a $(n_r-N)\times N$ matrix full of 0. $\boldsymbol{\Omega}_{N\times(N-1)}$ is generated through the algorithm described in the appendix of Nerger et al. (2012). The matrix $\boldsymbol{\Omega}_{N\times(N-1)}$ is designed so that the sample mean of the starting ensemble is equal to $\boldsymbol{x}_1^a$ and its sample covariance matrix is equal to matrix $\mathbf{P}_1^a$ reduced to its $N$ largest eigenvalues.

This method of generating the starting ensemble takes advantage of the extensive knowledge we have on the background statistics of the model. Several other methods have been tested to generate a starting ensemble, such as starting with random decorrelated snapshots of mantle convection simulations, second order exact sampling from $\boldsymbol{x}_1^f$ and $\mathbf{P}_1^f$, and several assimilations of the first observations $\boldsymbol{y}_1^o$. These alternative solutions resulted in reconstructions with larger initial errors and slower error decrease throughout the assimilation window, if any.

## 3.2 Forecast

Between time steps $k-1$ and $k$, the forward numerical code STAGYY computes independently the evolution of each of the analyzed states $\{\boldsymbol{x}_{k-1,n}^a\}_{n\in[1,N]}$ to produce a forecast ensemble $\{\boldsymbol{x}_{k,n}^f\}_{n\in[1,N]}$.

The forecast state is the average of the ensemble

$$\boldsymbol{x}_k^f = \frac{1}{N}\mathbf{X}_k^f\mathbf{1}_N. \tag{30}$$

The forecast error covariance matrix is given by the sample covariance matrix of the ensemble of forecast states

$$\mathbf{P}_k^f = \frac{1}{N-1}\mathbf{X}_k^f\left(\mathbf{I}_N - \frac{1}{N}\mathbf{1}_N\mathbf{1}_N^T\right)\left(\mathbf{I}_N - \frac{1}{N}\mathbf{1}_N\mathbf{1}_N^T\right)^T\mathbf{X}_k^{fT} \tag{31}$$

where $\mathbf{I}_N$ is the identity matrix of dimension $N \times N$. After several assimilation cycles, the finite size of the ensemble induces the underestimation of the error variance (van Leeuwen, 1999), and can lead to filter divergence. We observed this behavior in our case, and to stabilize the filter we apply covariance inflation, as suggested in Anderson and Anderson (1999) and Hamill et al. (2001).

We correct the forecast ensemble variance with an inflation factor $\gamma$ according to

$$\mathbf{X}_k^f \leftarrow \frac{1}{N}\mathbf{X}_k^f\mathbf{1}_N\mathbf{1}_N^T + \left[\mathbf{X}_k^f\left(\mathbf{I}_N - \frac{1}{N}\mathbf{1}_N\mathbf{1}_N^T\right)\right]\sqrt{\gamma}, \tag{32}$$

where $\leftarrow$ means that we replace the matrix on the left-hand side by the term on the right-hand side. $\gamma$ is computed following the same principles as in the suboptimal Kalman Filter developed in Bocher et al. (2016), i.e. by comparing the error on observations and the standard deviation of the innovation $\boldsymbol{d}_k$ defined as

$$\boldsymbol{d}_k = \boldsymbol{y}_k^o - \frac{1}{N}\mathbf{H}\mathbf{X}_k^f\mathbf{1}_N. \tag{33}$$

The inflation factor is

$$\gamma = \frac{V^d - V^o}{V^f}, \tag{34}$$

with

$$V^d = \mathrm{Tr}\left(\boldsymbol{d}_k\boldsymbol{d}_k^T\right), \tag{35}$$

$$V^o = \mathrm{Tr}(\mathbf{R}_k), \tag{36}$$

$$V^f = \mathrm{Tr}\left[\mathbf{H}\mathbf{X}_k^f\left(\mathbf{I}_N - \frac{1}{N}\mathbf{1}_N\mathbf{1}_N^T\right)\left(\mathbf{I}_N - \frac{1}{N}\mathbf{1}_N\mathbf{1}_N^T\right)^T\mathbf{X}_k^{fT}\mathbf{H}^T\right], \tag{37}$$

where $\mathrm{Tr}(\cdot)$ means the trace. The inflation factor is then truncated between a minimum value of 1 (to prevent further contraction of the ensemble spread) and a maximum value of $\gamma^+ = 1.25$ (to prevent overspread). Several values of maximum inflation factor have been tested, from $\gamma^+ = 1.1$ to $\gamma^+ = 2$, and showed little impact on the efficiency of the assimilation. A constant inflation factor was also tested, but the results with an adaptive inflation factor were substantially more accurate, especially for the first assimilation times.

## 3.3 Analysis

The analyzed state $\boldsymbol{x}_{kn}^a$ of the $n$th member of the ensemble is

$$\boldsymbol{x}_{kn}^a = \boldsymbol{x}_{kn}^f + \mathbf{K}_k \left( \boldsymbol{y}_{kn}^o - \mathbf{H}\boldsymbol{x}_{kn}^f \right) \tag{38}$$

where $\mathbf{K}_k$ is the Kalman Gain. $\boldsymbol{y}_{kn}^o$ is the observed data vector $\boldsymbol{y}_k^o$ to which a random perturbation of zero expected value and covariance matrix $\mathbf{R}_k$ is added, as is recommended in Burgers et al. (1998).

The Kalman Gain is defined as

$$\mathbf{K}_k = (\mathbf{P}_k^f \circ \mathbf{C})\mathbf{H}^T \left[ \mathbf{H}(\mathbf{P}_k^f \circ \mathbf{C})\mathbf{H}^T + \mathbf{R}_k \right]^{-1}, \tag{39}$$

where the matrix $\mathbf{P}_k^f$ is the sample covariance matrix of the ensemble of forecast states $\{\boldsymbol{x}_{kn}^f\}_{n\in[1,N]}$. We use a limited ensemble size (maximum 768) to estimate $\mathbf{P}_k^f$. Spurious correlations ensue, especially between distant points. To mitigate this effect, we implement localization directly on the forecast error covariance matrix by Schur multiplying (symbol $\circ$) $\mathbf{P}_k^f$ by the localization matrix $\mathbf{C}$, as introduced by Hamill et al. (2001) and Houtekamer and Mitchell (2001). The matrix $\mathbf{C}$ is itself the Schur product of a vertical localization matrix $\mathbf{C}_v$ and a horizontal localization matrix $\mathbf{C}_h$. The value of $\mathbf{C}_v(i,j)$ depends on the absolute radius difference of the $i-th$ and the $j-th$ components of the state vector and on the vertical correlation length $\ell_v$. The value of $\mathbf{C}_h(i,j)$ depends on the absolute angle difference of the $i-th$ and the $j-th$ components of the state vector and on the vertical correlation length $\ell_h$. Both values follow a Gaspari-Cohn compactly supported fifth-order piecewise rational function (similar to a Gaussian but with a compact support, Eq. (4.10) of Gaspari and Cohn, 1999).

We also tested the domain localization strategy as described in Janjic et al. (2011), since it is in some cases computationally more efficient and already implemented in PDAF. However, it led to a systematic failure of the assimilation. This is due to the nature of our problem: all the observations are located at the surface of the model and we aim at estimating the temperature field over the whole depth of the mantle. A vertical localization is as necessary as a horizontal localization, hence the localization has to be done directly on the forecast error covariance matrix and not only in the data space.

## 3.4 Implementation of the ensemble Kalman filter

We used the software environment PDAF (Nerger et al., 2005; Nerger and Hiller, 2013) in combination with the mantle convection code STAGYY (Tackley, 2008) to develop an ensemble Kalman filter code for mantle convection . PDAF provides a set of core routines computing in parallel the analysis steps for a range of ensemble based data assimilation techniques. It provides as well a set of standard routines to adapt the parallelization of a preexisting parallel forward numerical model and integrate the data assimilation routines. The final product is a highly scalable ensemble data assimilation code running both forecasts and analyses in parallel.

We modified the STAGYY code following the procedure recommended by PDAF (see the online documentation wiki at Nerger, 2016). We also made a few modifications in PDAF routines to allow for localization directly on the forecast error covariance with the ensemble Kalman filter. Additionally, we designed a basic observation database so as to load in a single step all the observations used in the data assimilation procedure.

## 4 A posteriori evaluation of the ensemble Kalman filter method

We test the data assimilation scheme on twin experiments using the model described in Sect. 2.1. Throughout this section, we compare the results of the ensemble Kalman filter for mantle circulation reconstructions to the results computed using the method developed in Bocher et al. (2016), hereafter referred to as method 1.

After describing the setup used for twin experiments, we test the robustness of the EnKF method and compare it to that of method 1. Then, we determine the range of data assimilation parameters which are suitable to conduct an ensemble data assimilation. Finally, we assess the ability of the scheme to actually reconstruct specific geodynamic structures.

### 4.1 Twin experiment setup

Twin experiments are a way to assess the accuracy of a data assimilation procedure in a controlled environment, where the true evolution is perfectly known.

First, we compute a reference state evolution using the forward numerical model, considered as the true state evolution, from which we extract the set of true state vectors $\{\boldsymbol{x}_k^t\}_{k \in [1,K]}$. Here, the timespan of the state evolution is 150 Myr and we sample true state vectors every 10 Myr. From these state vectors, we compute a time series of surface heat fluxes and surface velocities, following Eq. (12). We add to these observations a random Gaussian noise of standard deviation 10% of the root mean square of surface heat flux $q_{rms}$ and surface velocities $v_{rms}$ (we compute $q_{rms}$ and $v_{rms}$ from a free run of the dynamical model, they represent long term averages and are characteristic of the system dynamics). We obtain the time series of observations to assimilate $\{\boldsymbol{y}_k^o\}_{k \in [1,K]}$. It follows that the observation error covariance matrix $\mathbf{R}$ is diagonal and time independent.

Then, we perform ensemble data assimilation for the data set $\{\boldsymbol{y}_k^o\}_{k \in [1,K]}$, with the observation error covariance matrix $\mathbf{R}$. We did not consider any model error in the filter we describe, so the parameters of the model used in the data assimilation realizations are the same as those of the reference model.

We present here tests with different assimilation parameters, varying the number of members $N$, the vertical correlation length $\ell_v$ and the horizontal correlation angle $\ell_h$. Table 3 details the range of parameters tested.

We compute four different state evolutions to test the accuracy of the ensemble Kalman filter for different dynamical cases (the four state evolutions are described in the next section). Figure 3 shows the initial and final states of these evolutions, together with the result of global error evolution, and will be discussed in the next section.

### 4.2 Robustness of the assimilation algorithm

The evolutions of the global errors on the estimated temperature field and surface horizontal velocity field over the time period $\{1, ..., K\}$ are

$$\left[\epsilon_T^f(1), \epsilon_T^a(1), \epsilon_T^f(2), ..., \epsilon_T^f(K), \epsilon_T^a(K)\right] \qquad \text{and} \qquad \left[\epsilon_{u_\phi}^f(1), \epsilon_{u_\phi}^a(1), \epsilon_{u_\phi}^f(2), ..., \epsilon_{u_\phi}^f(K), \epsilon_{u_\phi}^a(K)\right], \qquad (40)$$

respectively, where $\epsilon_T^e(k)$ and $\epsilon_{u_\phi}^e(k)$, $e$ standing for $a$ (analysis) or $f$ (forecast) are

$$\epsilon_T^e(k) = \sqrt{\frac{\sum\limits_{l=1}^{L}\sum\limits_{m=1}^{M}\left(\overline{T}_k^e(\phi_l,r_m) - T_k^t(\phi_l,r_m)\right)^2 \mathcal{V}(\phi_l,r_m)}{\sum\limits_{l=1}^{L}\sum\limits_{m=1}^{M}\mathcal{V}(\phi_l,r_m)}} \quad \text{and} \quad \epsilon_{u_\phi}^e(k) = \sqrt{\frac{\sum\limits_{l=1}^{L}\left(\overline{u}_{\phi k}^e(\phi_l) - u_{\phi k}^t(\phi_l)\right)^2 \mathcal{V}(\phi_l,r_a)}{\sum\limits_{l=1}^{L}\mathcal{V}(\phi_l,r_a)}}$$

(41)

with $\mathcal{V}(\phi_l,r_m)$ the volume of the grid cell at longitude $\phi_l$ and radius $r_m$, $\overline{T}_k^e(\phi_l,r_m)$ the average temperature and $\overline{u}_{\phi k}^e(\phi_l)$ the average horizontal velocity of the estimated ensemble (either forecast or analysis) at longitude $\phi_l$ and radius $r_m$ and $r_a$, and where the superscript $t$ still refers to the true state.

We test the EnKF on one evolution, with sizes of the ensemble $N = 96$, $288$ and $768$ and for each combination of the following values of the data assimilation parameters: vertical correlation length $\ell_v = 0.3$, $0.5$, $0.7$ and $1$ and horizontal correlation angle $\ell_h = \pi/10$, $\pi/8$, $\pi/6$, $\pi/4$ and $\pi/2$ . We show in Fig. 1, for each ensemble size, the maximum and minimum values of errors on temperature (Fig. 1(a-c)) and on surface horizontal velocity (Fig. 1(d-e)), obtained for all these parameters, as a function of time. We also represent the background error on temperature $\epsilon_T^b(k)$ and on surface horizontal velocity $\epsilon_{u_\phi}^b(k)$

$$\epsilon_T^b(k) = \sqrt{\frac{\sum\limits_{l=1}^{L}\sum\limits_{m=1}^{M}(T^b(r_m) - T_k^t(\phi_l,r_m))^2 \mathcal{V}(\phi_l,r_m)}{\sum\limits_{l=1}^{L}\sum\limits_{m=1}^{M}\mathcal{V}(\phi_l,r_m)}} \quad \text{and} \quad \epsilon_{u_\phi}^b(k) = \sqrt{\frac{\sum\limits_{l=1}^{L}\left(u_\phi^b(r_a) - u_{\phi k}^t(\phi_l,r_a)\right)^2 \mathcal{V}(\phi_l,r_a)}{\sum\limits_{l=1}^{L}\mathcal{V}(\phi_l,r_a)}}$$

(42)

where $T^b$ and $u_\phi^b$ are 1D profiles corresponding to the average temperature and horizontal velocity, respectively, computed from a long run.

We choose the average error on temperature after analysis

$$\overline{\epsilon}_T^a = \frac{1}{K}\sum_{k=1}^{K}\epsilon_T^a(k)$$

(43)

as the global measure for the quality of the assimilation. For each ensemble size, the error evolution of the best assimilation (in the sense of minimum $\overline{\epsilon}_T^a$) is also shown in Fig. 1.

The error evolutions for temperature and surface horizontal velocity follow the analysis-forecast sequence: at each analysis time (every 10 Myrs), the error decreases abruptly, and during the forecast phases, the error increases.

For the surface horizontal velocity (Fig. 1(d-f)), the error evolutions are very similar regardless of the data assimilation parameters: the error decreases drastically during the analysis to a value of 25 to 50, while the amplitude of the error growth during the forecast phase evolves from around 200 for the first forecasts to around 100 at the end of the assimilation.

On the contrary, the evolution of the error on the temperature depends on the parameters of the assimilation. Fig. 1(a-c) shows that, for any size of the ensemble, it is possible to find a set of parameters leading to a drastic reduction of the global error on the temperature field after a few analyses. This first phase, when errors decrease quickly, lasts approximately 70 Myr,

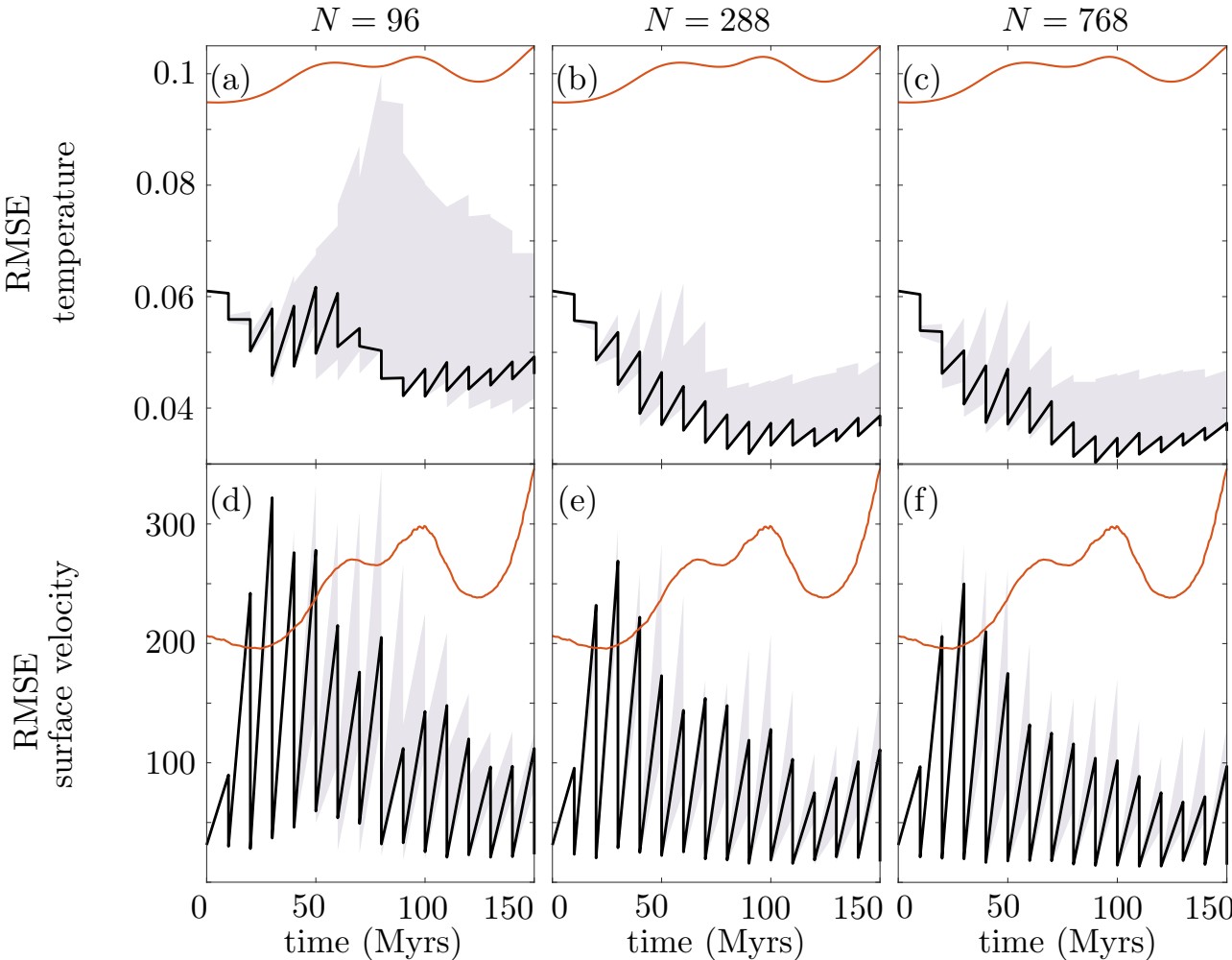

**Figure 1.** Time evolution of the errors on the estimated temperature field (panels (a) to (c)) and the estimated surface velocities (panels from (d) to (f)) obtained from data asssimilations with the same 150 Myr observation dataset, but different assimilation parameters. The size of the ensemble is $N = 96$ for (a) and (d), $N = 288$ for (b) and (e) and $N = 768$ for (c) and (f). The assimilations are computed for any combination of data assimilation parameters: $\gamma^+ = 1.25$, $\ell_v = 0.3, 0.5, 0.7$ and $1$ and $\ell_h = \pi/10, \pi/8, \pi/6, \pi/4$ and $\pi/2$. The black line represent the evolution of the error for the assimilation with the minimum average error on the analyzed temperature field: $N = 96$, $\ell_z = 0.5$, $\ell_h = \pi/6$, and $\gamma^+ = 1.25$ for (a) and (d), $N = 288$, $\ell_z = 0.7$, $\ell_h = \pi/10$ and $\gamma^+ = 1.25$ for (b) and (e), $N = 768$, $\ell_z = 0.5$, $\ell_h = \pi/4$ and $\gamma^+ = 1.25$ for (c) and (f). The gray area is delimited by the maximum and minimum values of errors at each time, for all data assimilations. The background error is represented in red, for reference.

which corresponds to the transit time of the dynamic system. After this phase, the error on temperature slowly increases with time, while remaining well below the errors measured for the first analyses. We can see that for $N = 288$ and $N = 768$, any combination of vertical and horizontal correlation lengths leads to errors lower than the first analysis. However, the difference between the maximum and the minimum errors obtained is greater than 0.01, which is large considering the background error

is only around 0.1. The best error evolutions for $N = 288$ and $N = 768$ are very similar, with a minimum error of 0.0318 and 0.0302 after 90 Myr, and an average global error after analysis of 0.0391 and 0.0378, respectively. During the assimilation of a dataset, most of the computational time is dedicated to the forecast step, so the data assimilation with 768 members is 2.7 times more expensive (computationally speaking) than the assimilation with 288 members. Since we obtain very similar results for $N = 288$ and $N = 768$, we favor the assimilation with 288 members.

We compute the error on the estimated temperature by comparing it to the true temperature field. However, in a realistic case, the true temperature is not known, and the evaluation of the data assimilation algorithm is based on the study of the statistics of the innovation vector $\boldsymbol{d}_k$ at forecast number $k$

$$\boldsymbol{d}_k = \boldsymbol{y}_k^o - \mathbf{H}\boldsymbol{x}_k^f. \tag{44}$$

After each forecast and just before analysis, we compute the Euclidean norm of the instantaneous innovation $d_k^i$ and the

Euclidean norm of the cumulative mean innovation $d_k^c$

$$d_k^i = \| \boldsymbol{d}_k \| \qquad \text{and} \qquad d_k^c = \left\| \frac{1}{k} \sum_{i=1}^{k} \boldsymbol{d}_i \right\| \tag{45}$$

Before computing these norms, we normalize the part of the innovation corresponding to surface heat flux and velocities by their respective root mean square values $q_{rms}$ and $v_{rms}$ (corresponding to time averages, characteristic of the dynamic system we are studying).

Figure 2 shows the evolution of $d_k^i$ and $d_k^c$ as a function of the number of forecasts for data assimilations with different sizes of ensemble and their respective optimum vertical and horizontal correlation lengths.

The evolution of the cumulative mean of the innovation $d_k^c$ allows us to check some aspects of the consistency of the data assimilation algorithm. Indeed, the derivation of the EnKF equations assumes that the error on observations $\boldsymbol{y}^o$ and the error on the forecast data $\mathbf{H}\boldsymbol{x}^f$ are unbiased. Such hypotheses imply that the statistically expected value of $\boldsymbol{d}$ is zero, which means

that the norm of the cumulative innovation should converge to zero as the number of forecasts increases. Figure 2(a) shows the cumulative innovation constantly decreasing throughout the assimilation, with comparable values for $N = 288$ and $N = 768$, and slightly higher values for $N = 96$.

The norm of the instantaneous innovation $d_k^i$ measures the distance between the forecast data and the observation, and therefore allows us to monitor the success (or failure) of the assimilation. In Fig. 2(b), we can see that the norm of the instantaneous

innovation decreases during the first 8 forecasts, i.e. 70 My, and then oscillates for the rest of the assimilation. The comparison of Fig. 1 and Fig. 2 reveals one important pitfall of the application of data assimilation to our problem. After the 10th assimilation, the instantaneous innovation (Fig. 2(b)) is almost the same for $N = 96$, $N = 288$ and $N = 768$, while the global error on the estimated temperature field (Fig. 1) is clearly higher for $N = 96$ than for $N = 288$ or 768. This is because the

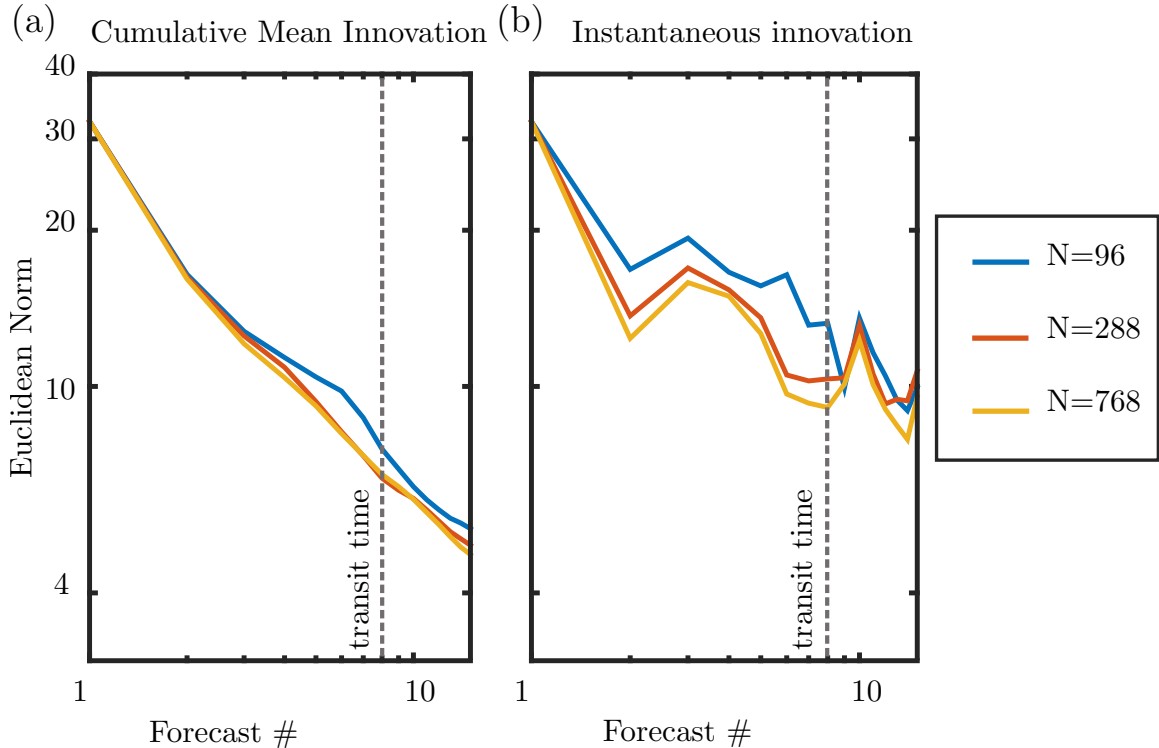

**Figure 2.** Evolution of (a) the cumulative mean innovation and (b) the norm of the instantaneous innovation, as a function of the number of forecasts performed, and for different ensemble sizes. For each size of the ensemble, the evolutions correspond to the best combinations of correlation length parameters: $N = 96$, $\ell_z = 0.5$, $\ell_h = \pi/6$ and $\gamma^+ = 1.25$ ; $N = 288$, $\ell_z = 0.7$, $\ell_h = \pi/10$ and $\gamma^+ = 1.25$ and $N = 768$, $\ell_z = 0.5$, $\ell_h = \pi/4$ and $\gamma^+ = 1.25$.

instantaneous innovation measures the distance between observed and forecast data at the surface, while the error measures the distance between the estimated and true temperature field, not only at the surface but also in depth. This means that for a same innovation at the surface, the error on the temperature field at depth can vary substantially. In other words, the instantaneous innovation does not necessarily vary the same way the true error on the temperature field does.

5    We also tested the assimilation algorithm for 4 different state evolutions, with the optimal parameters for an ensemble size of $N = 288$ members ($\ell_v = 0.7$ and $\ell_h = \pi/10$). Figure 3 shows the initial and final temperature fields of the evolutions, together with the evolution of the global error, the spread of the ensemble, the background error and the error evolution using method 1.

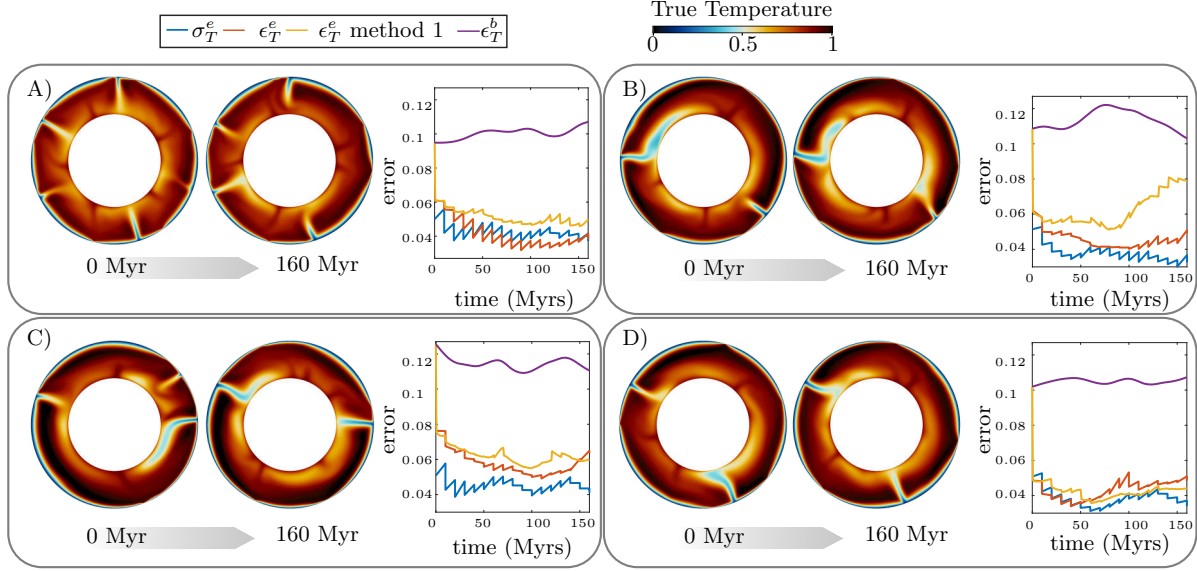

**Figure 3.** Evolution of the error ($\epsilon_T^e$, red) as a function of time for 4 different evolutions with $N = 288$, $\gamma^+ = 1.25$, $\ell_v = 0.7$ and $\ell_h = \pi/10$, compared to the evolution of the spread of the ensemble ($\sigma_T^e$, blue), the evolution of the error with the technique of Bocher et al. (2016) ($\epsilon_T^e$ method 1, yellow) and the background error ($\epsilon_T^b$, purple). The initial and final states of the true evolutions are represented on the left of each corresponding graph.

The spread of the ensemble is an estimation of the uncertainty on the state. We compare the evolution of $\epsilon_T^e$ to the global standard deviation of the temperature field of the ensemble:

$$\left[ \sigma_T^f(1), \sigma_T^a(1), \sigma_T^f(2), ..., \sigma_T^f(K), \sigma_T^a(K) \right] \tag{46}$$

with $\sigma_T^e(k)$ defined as

$$\sigma_T^e(k) = \sqrt{ \frac{\sum\limits_{n=1}^{N} \sum\limits_{l=1}^{L} \sum\limits_{m=1}^{M} \left( T_{kn}^e(\phi_l, r_m) - \overline{T}_k^e(\phi_l, r_m) \right)^2 \mathcal{V}(\phi_l, r_m)}{(N-1) \sum\limits_{l=1}^{L} \sum\limits_{m=1}^{M} \mathcal{V}(\phi_l, r_m)} }. \tag{47}$$

We compute the error for an estimated state evolution with method 1 using Eq. (42).

Although we computed the four state evolutions using the same forward modeling code and with the same values of physical parameters (as described in Table 1), they show different geodynamic configurations: Evolution A has a shorter wavelength of convection, with the persistence of 4 subductions, 3 ridges and 5 upwellings, the death of one ridge and creation of two. Evolutions B, C and D have longer wavelengths of convection, with two major downwellings, stable throughout the evolutions.

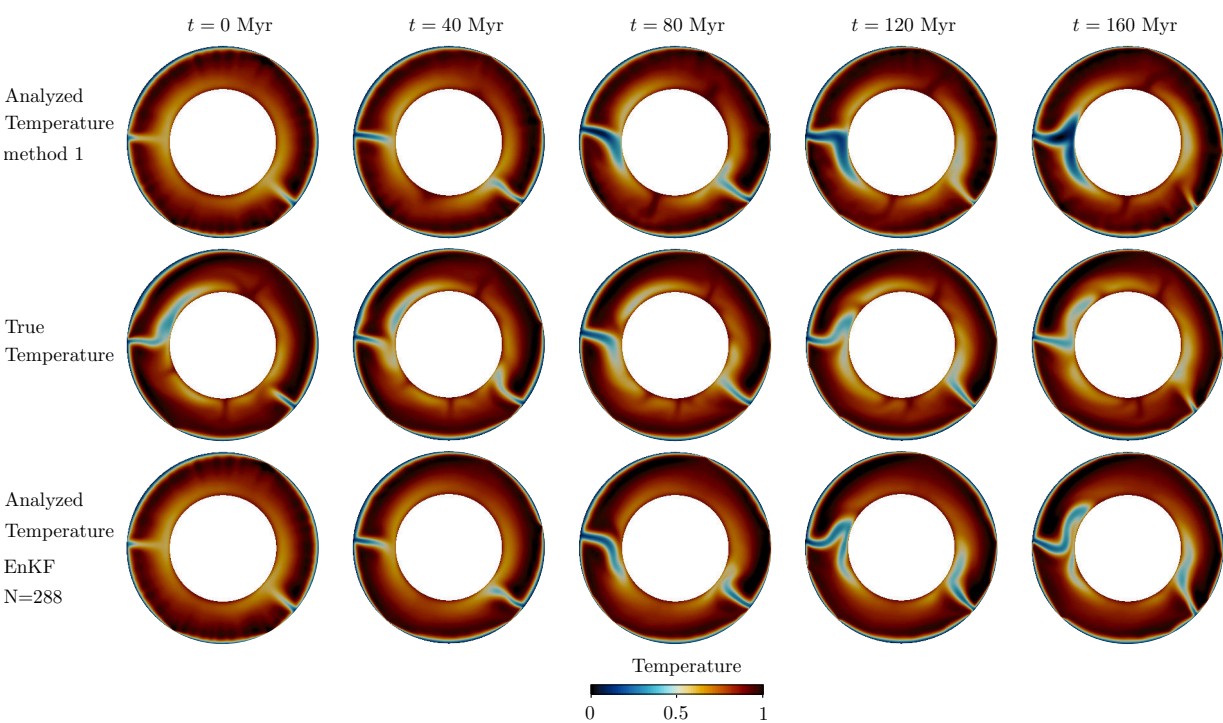

**Figure 4.** Comparison of temperature field evolutions for evolution B. The first row depicts the evolution of the analyzed temperature field with method 1. The second row is the true evolution of the temperature field. The third row is the evolution of the analyzed temperature field with ensemble Kalman filter, $N = 288$, $\ell_v = 0.7$ and $\ell_h = \pi/10$.

In evolution B, one of these downwellings has a very large negative temperature anomaly at the bottom of the domain. In evolution C, the remnant of a subduction merges with a larger subduction into a single downwelling.

In the 4 cases, the errors on the estimated temperature field systematically decrease during the analysis step for the EnKF algorithm. The errors stay below the first analysis error for evolutions A, B and C, while they reach slightly higher values for evolution D. The error of the EnKF is always lower than that obtained with method 1 for the first $50$ My. The average error is lower for the EnKF than for method 1 in 3 out of 4 cases. The average standard deviation of the ensemble (ensemble spread) is of the same order of magnitude as the true error. However, its evolution is not the same as the true error, with differences between both of more than $0.02$ for some part of evolution C, for example. Moreover, in three out of four cases (cases B, C and D), the spread of the ensemble is much lower than the true error. For evolutions C and D, the results of the two methods are comparable whereas the assimilation with EnKF performs better than method 1 for evolutions A and B.

For evolution B, method 1 fails to reconstruct accurately the evolution, with the error reaching values greater than $0.08$ at the end of the assimilation. This case is further investigated on Fig. 4 and 5. Figure 4 compares the true temperature field evolution

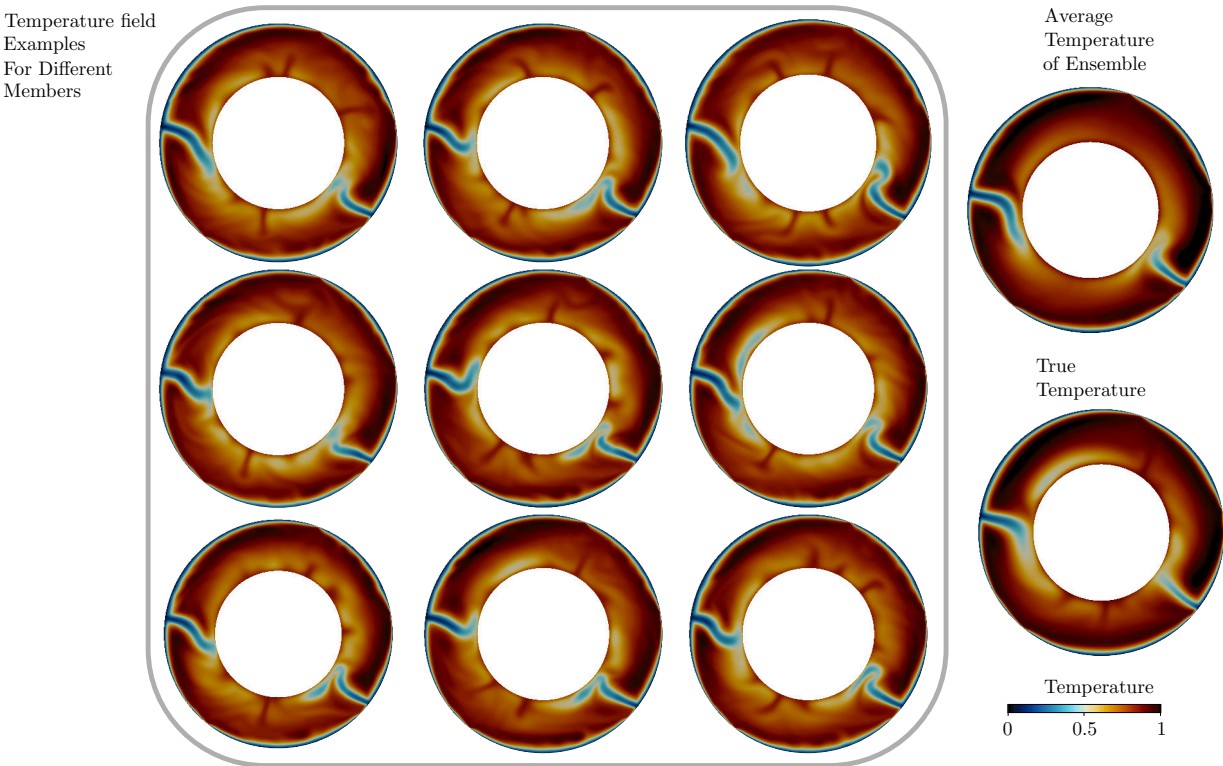

**Figure 5.** Example of temperature fields of the members of the ensemble. This example is taken after 80 Myr, for the assimilation of evolution B, with ENKF $N = 288$, $\ell_v = 0.7$ and $\ell_h = \pi/10$.

with the analyzed temperature field of method 1 and of the ensemble Kalman filter with $N = 288$, $\ell_v = 0.7$ and $\ell_h = \pi/10$. The sudden increase in the error of the estimated temperature field for method 1 seen on Fig. 3(b) happens after around 80 Myr of assimilation, when the direction of bending at the bottom of the domain changes for the downwelling on the left side (see Fig. 4, second row). The analyzed temperature field of Method 1 does not predict this change of direction (see Fig. 4, first row),

5   while the analyzed temperature field of the ENKF predicts it (see Fig. 4, third row). Method 1 computes only the evolution of the best estimate of the system. The computation of only one estimate ignores that, in this case, a slight perturbation of the estimated state could lead to a totally different dynamics. On the contrary, the EnKF method computes the evolution of an ensemble of perturbed solutions and thus takes into account the nonlinearity of the solution, at least for the forecast stage. Figure 5 shows examples of the analyzed temperature fields of different ensemble members for evolution B, after 80 Myr of

10   assimilation. Although the average temperature fields displays a downwelling bending to the right, the ensemble members show a wide variety of downwelling geometries.

### 4.3 Reliability of the ensemble

On Fig. 3, the standard deviation of the temperature of the ensemble, $\sigma_T^e$, is lower than the error on temperature, $\epsilon_T^e$, for some state evolutions. This indicates that we cannot rely on the spread of the ensemble to estimate accurately the evolution of the global error on the temperature field. To investigate the reliability of the ensemble in more details, we compute rank histograms for surface heat flux and velocity (Fig. 6), and for the temperature at the surface, mid-domain and at the bottom (Fig. 7).

Rank histograms were first described independently by Anderson (1996), Hamill and Colucci (1996, 1997), and Talagrand (Harrison et al., 1995; Talagrand et al., 1997). They are a tool to diagnose systematic biases and misestimations of the uncertainty in an ensemble of forecasts (Hamill, 2001). To obtain the rank histograms of Fig. 6 and 7, we proceed as follows.

1. Selection of the variable and the verification. We compute rank histograms for surface heat fluxes (Fig. 6(a,c,e)), surface velocities (Fig. 6(b,d,f)), and surface, mid-mantle and bottom temperature (Fig. 7). For Fig. 6(a,b) and Fig. 7, the ensemble is checked against the true value while for Fig. 6(c,d,e,f), it is checked against the observed value. In this context, the true values are the verification for Fig. 6(a,b) and Fig. 7, and the observed values are the verification for Fig. 6(c,d,e,f), respectively.

2. Selection of the sampling points. To be able to interpret our rank histograms, we need to populate them with samples that are independent. To do so, we use the four evolutions presented in figure 2, and, for each evolution, we select points that are spaced from each others by the correlation angle $\ell_h = \pi/10$, and taken after 10, 80 and 150 Myrs of assimilation. We obtain 120 sampling points per histogram.

3. Determination of the rank of the verification. At each sampling point, we determine the rank of the verification in a vector composed of all the values taken by the ensemble plus the verification, in ascending order.

4. Computation of the rank histogram. In order to have bins of constant width, we choose 17 ranks as the bin width ($289 = 17^2$).

If the ensemble statistics is reliable, then the true value of a given variable and the values of the ensemble of forecasts can be considered as random draws from the same distribution. In this hypothesis, the rank of the true value follows a uniform law, and the rank histogram should be flat. We represented the expected rank counts for a flat histogram as a dashed line in Fig. 6 and 7. If this is not the case, the shape of the rank histogram provides indications on the existence of biases and under- or over-dispersion of the ensemble (even though the shape of a rank histogram can also be affected by other factors, see e.g. Hamill, 2001).

To guide our interpretations, we perform the $\chi^2$ goodness-of-fit-test (see e.g. Wilks, 2006, Sect. 5.2.5 and 7.7.2) to test if our rank histograms are significantly non uniform. We compute the value

$$\chi^2 = \sum_{i=1}^{17} \frac{(\#o_i - \#e_i)^2}{\#e_i} \tag{48}$$

where $\#o_i$ is the bin count in the $i-th$ bin and $\#e_i$ is the expected count for a uniform distribution $120/17 \approx 7.06$. The values of $\chi^2$ are written on each histogram of Fig. 6 and 7. If the ranks we sampled come from a uniform distribution, then

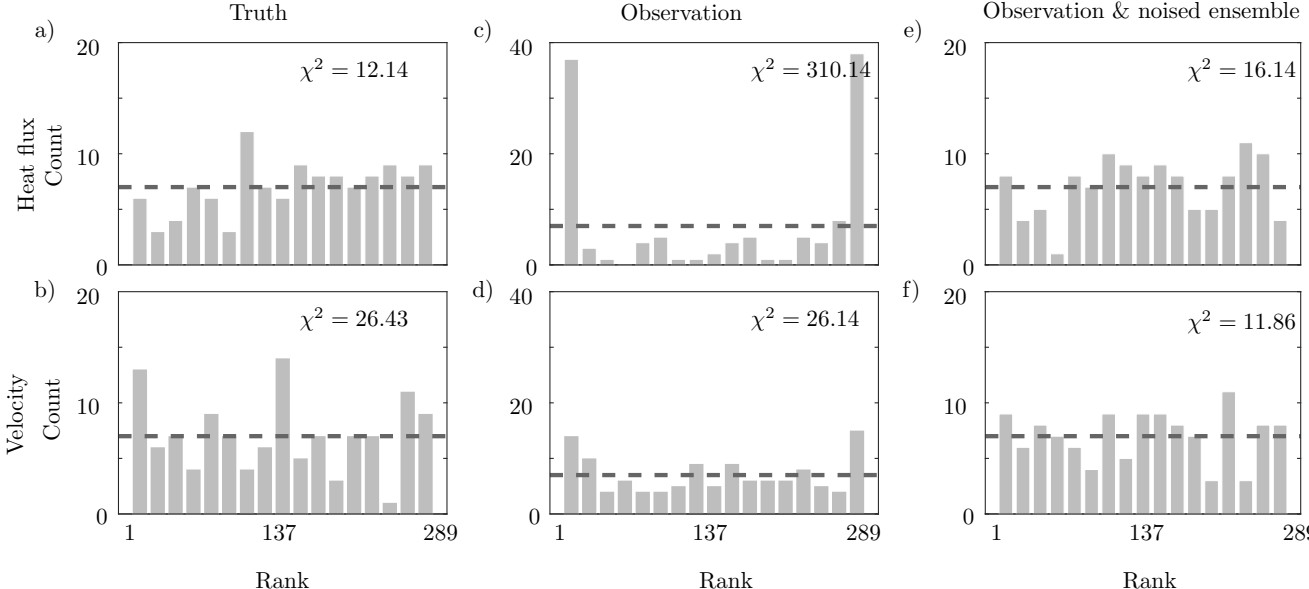

**Figure 6.** Rank histograms of the surface true heat flux (a) and velocity (b) as well as the surface observed heat flux (c) and velocity (d), computed from the 4 evolutions of figure 2. The dashed lines represent the count for each bin if the rank histograms were flat.

$\chi^2$ follows a chi-square probability law with $17 - 1 - 1 = 15$ degrees of freedom. In this hypothesis, the probability to obtain a $\chi^2 \geq \chi_c^2 = 24.996$ is 0.05. We take this value of $\chi_c^2$ as the critical value over which we consider that the rank histogram is significantly non-uniform.

The left column of Fig. 6 represents the rank histograms of the true surface heat flux, Fig. 6(a) and velocity Fig. 6(b).

Histogram 6(a) shows a slightly higher occurrence of the true heat flux in higher ranks within the ensemble. This would suggest that the ensemble estimation of surface heat flux is biased towards the lower values. However the $\chi^2$ value for histogram 6(a) is well below the critical value $\chi_c^2$, so that we cannot say that the rank histogram is significantly non-uniform. On the contrary, the rank histogram of surface velocities (Fig. 6(b)) has a $\chi^2 = 26.43 > \chi_c^2$: it is significantly non-uniform. This histogram is more populated in the bins corresponding to the lowest and highest ranks ($1 - 17$ and $255 - 289$). This would suggest ensemble

under-dispersion, even though the shape of the rank histogram is more complex than the classical U shape associated with ensemble under-dispersion (in particular, the midle ranks ($137 - 153$) are also highly populated).

In an assimilation with Earth data, the truth is not known, and we would have to draw rank histograms using observed data. The question is: would we come to the same conclusion about the reliability of the ensemble as with the histograms 6(a,b)? The middle column of Fig. 6 represents the rank histograms of the observed heat flux on Fig. 6(c) and observed velocity on

Fig. 6(d). Both histograms 6(c) and (d) have a distinct U shape, with $\chi^2$ of 310.14 and 26.14, respectively. For the surface heat flux, the difference between rank histograms of the truth and the observation is dramatic (the $\chi^2$ value jumps from 12.14 to 310.14). For the surface velocity, the difference is less striking, even though the U shape is much clearer on the rank histogram

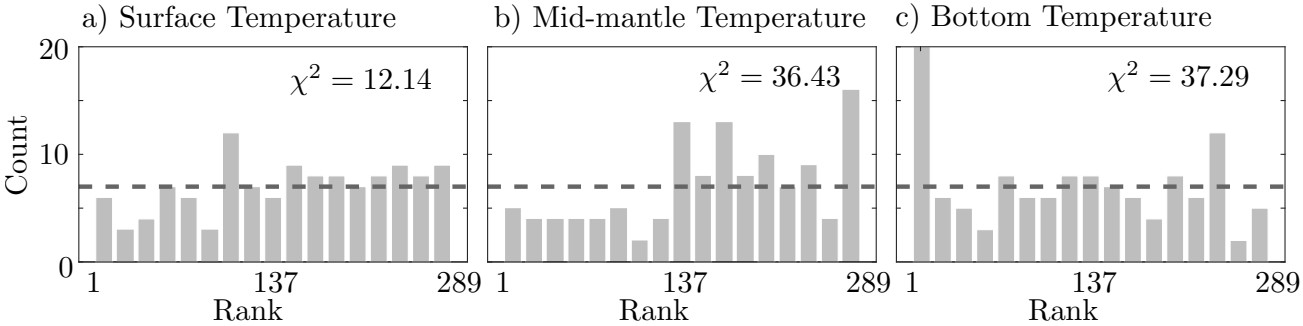

**Figure 7.** Rank histograms for temperature at the surface (a), mid-mantle (b) and at the bottom of the model (c), computed from the 4 evolutions of figure 2. The dashed lines represent the count for each bin if the rank histograms were flat.

of observed velocities.The more pronounced U shape for the rank histograms of both observed heat flux and velocity indicates that the ensemble is not as under-dispersed around the truth as what could be deduced from the rank histograms with noised observations. In other words, noise in the observation has a major effect on the shape of the rank histogram, so that we cannot interpret the reliability of the ensemble by looking directly at the rank histograms of the observations.

Since the noise in the observations largely affects the shape of the rank histograms, we need to add noise to the ensemble members before computing the rank histograms, as explained in Anderson (1996) and Hamill (2001). The noise we add to each ensemble member has the same standard deviation as the noise affecting the observed data. The right column of Fig. 6 represents such rank histograms for heat flux on Fig. 6(e) and velocity on Fig. 6(f). Both $\chi^2$ score are well below $\chi^2_c$: we cannot say that the rank histograms are significantly non-uniform. It is not possible to detect the under-dispersion of the ensemble for

surface velocity using only observed data.

Figure 7 shows the rank histograms for temperature at different depth. At the surface (Fig. 7(a)), the rank histogram is the same as the rank histogram for the true heat flux Fig. 6(a), since there is a linear relationship between surface temperature and surface heat flux. It follows that the rank histogram of surface temperature is not significantly non-uniform. At mid-mantle (Fig. 7(b)), the rank histogram of temperature is significantly non-uniform, with a $\chi^2 = 36.43 \geq \chi^2_c$. It is more populated

towards the higher values, which could indicate that the temperature ensemble in the mid-mantle is biased towards the lower values. At the bottom (Fig. 7(c)), the rank histogram of temperature is also significantly non-uniform, with a $\chi^2 = 37.29 \geq \chi^2_c$. The first bin of the rank histogram is highly populated, while the rest of the histogram is roughly flat. This suggests that the ensemble is biased towards the hotter temperatures at the bottom of the model.

In conclusion, Fig. 7 shows that the ensemble is reliable at the surface for temperature, but becomes unreliable at depth.

The lower value of the standard deviation $\sigma^e_T$ compared to the true error $\epsilon^e_T$ observed in Fig. 3 in 3 out of 4 cases is due to a misestimation of the error on the temperature at depth by the ensemble. We discuss this point in more details in Sect. 5.

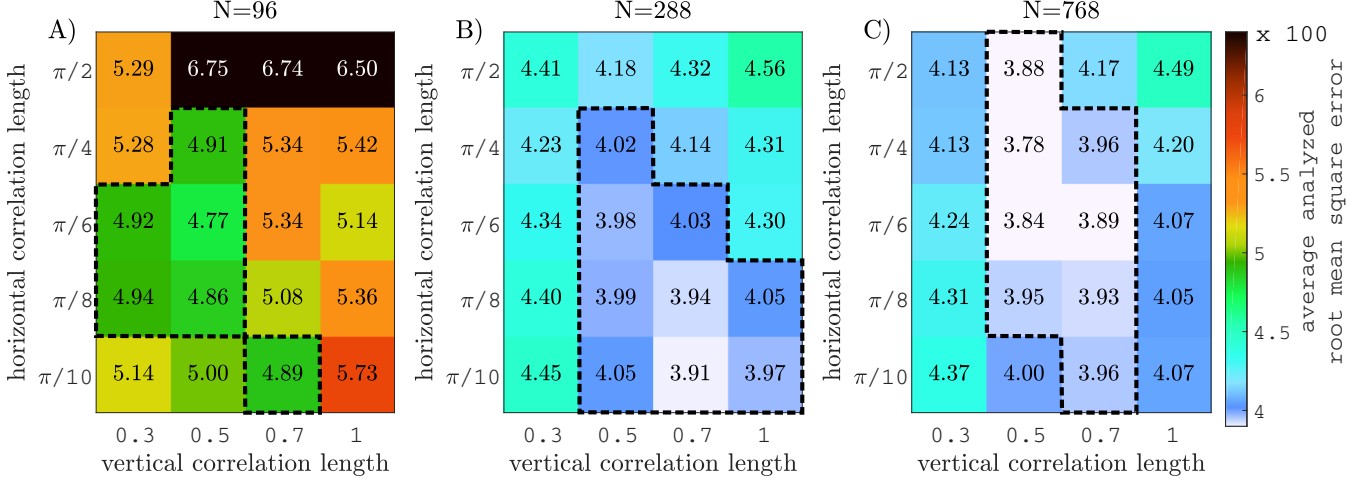

**Figure 8.** Value of the average analyzed error for assimilations performed using the dataset generated by evolution A of Fig. 3, with different sizes of the ensemble, and vertical and horizontal correlation lengths. (a) for 96 ensemble members, (b) for 288 ensemble members, (c) 768 ensemble members. The dashed lines delimit the zones for which errors are less than $\bar{\epsilon}^a_{Tmin}(N) + 0.002$.

## 4.4 Effect of the data assimilation parameters on the quality of the estimation

As shown in Fig. 1, the choice of $N$, $\ell_v$ and $\ell_h$ is critical to minimize errors in the assimilation, with errors on the estimated temperature field varying from 0.03 to more than 0.1 depending on the choice of parameters. We investigate further the effect of these parameters by comparing the average global errors after analyses, $\bar{\epsilon}^a_T$, for different combinations of $N$, $\ell_v$ and $\ell_h$.

Figure 8 displays the values of $\bar{\epsilon}^a_T$ for sizes of ensemble $N = 96$, 288 and 768 (Fig. 8(a), (b) and (c) respectively) with $\ell_v$ varying between 0.3 and 1, and $\ell_h$ between $\pi/10$ and $\pi/2$. As in Fig. 1, we observe a dichotomy between assimilations with $N = 96$ members, with higher errors, and assimilations with $N = 288$ and 768, with lower errors.

For each size of ensemble $N$ we identify the pair $(\ell_v, \ell_h)$ that leads to the assimilation with the lowest error $\bar{\epsilon}^a_{Tmin}(N)$. From this minimum value $\bar{\epsilon}^a_{Tmin}(N)$, we select all the pairs $(\ell_v, \ell_h)$ that lead to data assimilation with global errors less than

10 $\bar{\epsilon}^a_{Tmin}(N) + 0.002$. As the size of the ensemble increases, the optimal lengths of correlations $(\ell_v, \ell_h)$ tend to increase. This is a classical effect (Houtekamer and Mitchell, 1998), observed in ensemble Kalman filters for various dynamical systems. As $N$ increases, the amplitude of noise in the sample correlation matrix $\mathbf{P}^f$ decreases, and small, yet real, correlations between distant points can be taken into account (Hamill et al., 2001). Between ensemble sizes of $N = 96$ and $N = 288$ the zone of optimal correlations is displaced towards the greater vertical correlation lengths. When we increase the size of the ensemble

from $N = 288$ to $N = 768$, the zone of optimal correlations is displaced towards greater horizontal correlation angles. So the accurate estimation of correlations between points on the same vertical level needs less samples than between points on the same horizontal level. This is due to the specifics of mantle convection dynamics. The highly nonlinear rheology produces plates at the surface with values of velocity and temperature that may vary substantially (by one or two orders of magnitude) on short distances in the horizontal direction, especially because of pseudoplasticity. On the contrary, highly viscous cold

downwellings establish a strong continuity in the vertical direction. Given that a small perturbation can trigger the formation of a new plate boundary (see Sect. 2.1), those scales of variability reverberate through the ensemble covariance matrix.

For the ensemble size $N = 288$ and all the values of $(\ell_v, \ell_h)$, we additionally evaluate the average global ensemble spread

$$\overline{\sigma}_T^a = \frac{1}{K} \sum_{k=1}^{K} \sigma_T^a(k), \tag{49}$$

the average norm of the instantaneous innovation

$$\overline{d^i} = \frac{1}{K} \sum_{k=1}^{K} \| \boldsymbol{y}_k^o - \mathbf{H} \boldsymbol{x}_k^f \| \tag{50}$$

and the cumulative mean innovation after $K$ forecasts:

$$d_K^c = \left\| \frac{1}{K} \sum_{k=1}^{K} \left( \boldsymbol{y}_k^o - \mathbf{H} \boldsymbol{x}_k^f \right) \right\|. \tag{51}$$

These three values are indicators of the accuracy of the assimilation and can be computed in the case of an assimilation with
Earth data, unlike $\overline{\epsilon}_T^a$.

Figure 9 represents these results along with the true error $\overline{\epsilon}_T^a$. The ensemble of optimal data assimilation parameters is also outlined ($\overline{\epsilon}_T^a < \overline{\epsilon}_{Tmin}^a(N) + 0.002$).

Overall, the average ensemble spread $\overline{\sigma}_T^a$ (Fig. 9(b)) decreases when $\ell_h$ and $\ell_v$ increase, with a minimum for $\ell_h = \pi/2$ and $\ell_v = 1$. The higher the correlation lengths, the more covariances will be taken into account in the analysis, and the analyzed
members will be closer to each others and $\overline{\sigma}_T^a$ lower. The average ensemble spread $\overline{\sigma}_T^a$ is of the same order of magnitude as the true error $\overline{\epsilon}_T^a$. Moreover, there is a local minimum of $\overline{\sigma}_T^a$ at $\ell_v = 0.7$ and $\ell_h = \pi/10$. These parameters correspond to the minimum true error $\overline{\epsilon}_T^a$.

The average norm of instantaneous innovations and the norm of the cumulative innovations display the same behavior: they decrease with increasing vertical and horizontal correlation lengths. The longer the correlations lengths, the closer the forecast
data are to the observations, and the less biased the assimilation. This means that a better fit to the observations does not necessarily imply a better fit to the true temperature field. In a realistic context, the result of the assimilation should be checked against independent data to evaluate its accuracy. In the case of the Earth's mantle, independent data could be for example the geoid or tomographic models.

### 4.5 Accuracy of the reconstruction of geodynamic structures

We focus on three key flow structures: 1) downwelling slabs (subduction) 2) ridges, i.e. shallow structures resulting from divergent plates at the surface, 3) plumes, hot upwellings rising from the base of the model.

Figure 10 shows the final state of the assimilation after 150 My for the evolution A of Fig. 3. We selected 3 assimilations: EnKF96, an ensemble Kalman filter with $N = 96$, $\ell_v = 0.5$ and $\ell_h = \pi/6$ (first row), EnKF288 an EnKF with $N = 288$, $\ell_v = 0.7$ and $\ell_h = \pi/10$ (second row) and the assimilation with method 1 (third row). We do not show the ensemble Kalman

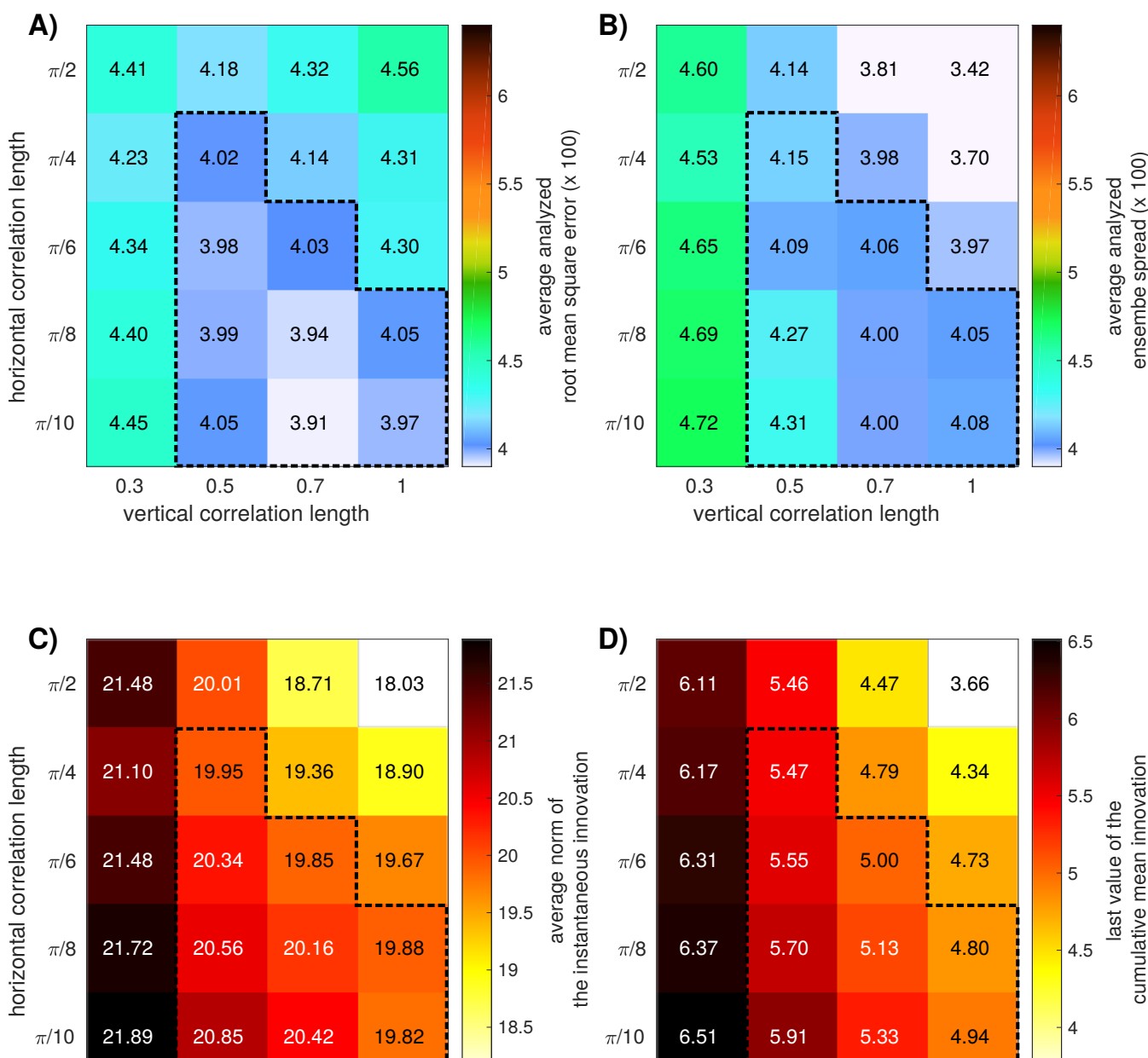

**Figure 9.** Value of (a) mean analyzed error, (b) mean ensemble spread, (c) norm of instantaneous innovation, (d) norm of cumulative innovation after $K = 16$ forecasts for $N = 288$, and different vertical and horizontal correlation lengths. The dashed line delimits the zone for which errors are less than $\bar{\epsilon}^a_{Tmin}(288) + 0.002$.

filter with 768 members since the resulting temperature field is almost indistinguishable from that of EnKF288. The first column represents the true temperature field, which is the same for all assimilations. The second column is the analyzed temperature field, i.e. the average of the temperature fields of the analyzed ensemble members. The third column is the absolute temperature error, and the fourth column is the standard deviation of the ensemble spread, which is an estimate of the error on the analyzed

temperature field.

Globally, the EnKF288 and EnKF96 solutions for the temperature field are smoother than the solution of method 1. We observe this difference especially in the asthenosphere, the part of the mantle below the top boundary layer. For method 1, the asthenosphere shows short wavelength temperature variations. These variations are absent from the true temperature field and are inconsistent with convection solutions with the chosen parameters. They stem from the amplification of the noise in the

observations during the analysis. Moreover, the asthenosphere of the analyzed temperature field of method 1 is hotter than the true temperature.

Both EnKF96 and EnKF288 reconstruct successfully the ridges locations and structures, as testified by their error fields. On the contrary, method 1 fails to reconstruct the ridge on the top right of the domain. It also predicts a ridge that does not exist in the true state (in the top left quadrant). On the right of the domain, another ridge is associated with a vertical positive

temperature anomaly underneath. This pattern is found regularly under ridges when applying method 1. This is due to the use of a constant forecast error covariance matrix, $\mathbf{P}_0^f$ for the analysis. This constant matrix does not take into account the specifics of the dynamics under a ridge, where the positive anomaly is generally shallow. We do not observe this detrimental effect in the EnKF assimilations, where we compute the forecast error covariance matrix $\mathbf{P}_k^f$ at each analysis time from the forecast ensemble.

All three assimilations reconstruct the subductions and predict accurately the bending direction of slabs at the base of the model. Method 1 tends to underestimate the amplitude of the negative temperature anomalies whereas both EnKF assimilations overestimate them. This is especially noteworthy for the bottom left subduction. Moreover, the estimated slabs are wider than the true slabs. However, we note two arguments in favor of the EnKF: first, the estimation of the slab improves when the size of the ensemble increases and second, the local standard deviations of the ensemble indicates that the estimation in this part of

the domain is less accurate.

Both EnKF288 and EnKF96 solutions do not show any plume at the base of the mantle. However, the ensemble spread shows a greater uncertainty on the places where plumes occur. Method 1 predicts the approximate location of all plumes, but their geometry is not accurate. Method 1 provides only one estimate of the temperature field. In this evolution, the plumes are allowed to develop. EnKF96 and EnKF288 provide an ensemble of states. Each state develops plumes at different locations

and their averages show only a slightly hotter anomaly over a wide area of possible location for the plumes, as we showed earlier in Fig. 5 for another assimilation.

To illustrate how different flow structures are reconstructed, we plot on Fig. 11 the time evolution of the EnKF288 ensemble surface, mid-domain and bottom temperature at the longitude of a) a subduction, b) a plume, c) a ridge initiation and d) a stable ridge. Figure 10 shows the location of these geodynamical features on the true temperature field. We plot the temperature

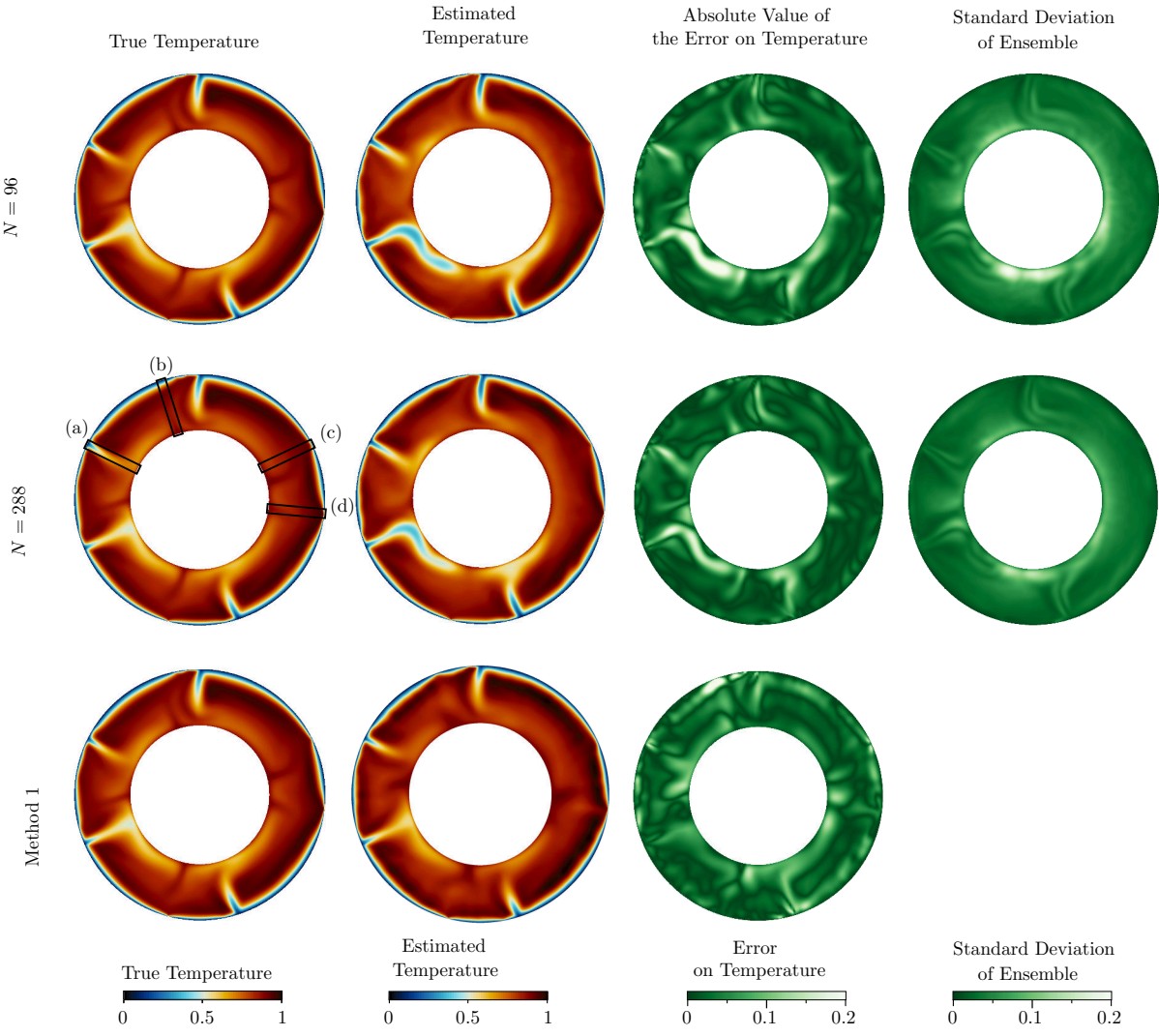

**Figure 10.** Comparison of estimated states after 150 Myr for the evolution A of Fig. 3. First row: ensemble Kalman filter with $N = 96$, $\ell_v = 0.5$ and $\ell_h = \pi/6$; second row: $N = 288$, $\ell_v = 0.7$ and $\ell_h = \pi/10$ (these localization length correspond to the optimal parameters determined previously.); third row for method 1 (third row). The first column represents the true temperature field at 150 Myr, the second column is the analyzed temperature field, the third is the absolute error on temperature value and the fourth is the estimated error on the analyzed field (spread of the ensemble). On the true temperature field of EnKF288 we framed the location of the subduction (a), plume (b), ridge initiation (c) and stable ridge (d) studied in Fig. 11.

evolutions at the surface, mid-mantle and at the bottom of the domain. Note that the surface and bottom values of temperature actually correspond to the values of the first points below the surface and above the bottom of the domain, respectively.

At the surface, the temperature is corrected accurately at each analysis, with a difference between the true temperature and the analyzed temperature of less than 0.01. The correction associated with the analysis gradually decreases with depth due to both covariance localization and the dynamics of the system.

For the subduction, the correction is first done on the surface, and then propagates gradually in depth. The reconstruction of mid-mantle temperature becomes accurate after 40 My, and at the bottom of the model after 70 My, which is the value of the transit time. At the surface, the spread of the ensemble decreases as more data are assimilated. On the contrary, the spread of the ensemble remains steady for mid-mantle depths and at the bottom of the domain. For these depths, only the average temperature varies.

At the surface for the plume, the spread of the ensemble is very low except for a peak at 40 My, which corresponds to an instability, corrected after one analysis. We note that this instability affects greatly method 1 since it leads to the false prediction of the ridge seen in Fig. 10. At mid-mantle, the ensemble average is slowly converging to the true temperature. At the bottom, the estimated temperature is lower than the true temperature, although it slightly increases throughout the assimilation.

The ridge initiation shows how new observations affect the spread of the ensemble. At the surface, the spread of the ensemble remains low until 100 My, the time of initiation of the ridge. From then on, the estimated temperature increases and the ensemble members follow the cycle of increasing spread during forecast and dramatic decrease of spread during analysis. The temperature in the mid-mantle is estimated with a very good accuracy after 50 My. On the contrary, the assimilation does not predict the evolution of the temperature at the bottom of the domain, although the true temperature falls within the zone defined by the standard deviation of the ensemble after 50 My.

For the stable ridge, the spread of the ensemble at the surface is increasing during forecast and decreasing dramatically during the analysis. At mid-mantle, the estimated temperature becomes accurate after 100 My. At the bottom of the domain the temperature is underestimated although it follows the variations of the true temperature: increase of temperature at the beginning of the assimilation and slight decrease at the end of the assimilation.

## 5   Discussion

We chose the ensemble Kalman filter method for its ease of implementation and flexibility to adapt to different forward numerical models. Indeed, as long as the nature of the state and observations does not change, the computation of the analysis step remains the same regardless of the convection code used. On the contrary, the alternative method, variational data assimilation, requires the development of an adjoint code that needs further development for each additional complexity added to the forward model (see Kalnay et al., 2007, for a comparison of EnKF and 4D variational methods). For the mantle circulation problem, this results in a series of derivation of the adjoint model considering different approximations (Ismail-Zadeh et al., 2003; Bunge et al., 2003; Ghelichkhan and Bunge, 2016; Worthen et al., 2014). The ability of a data assimilation scheme to adapt to different numerical codes is a particularly important issue for mantle convection since models are in constant evolu-

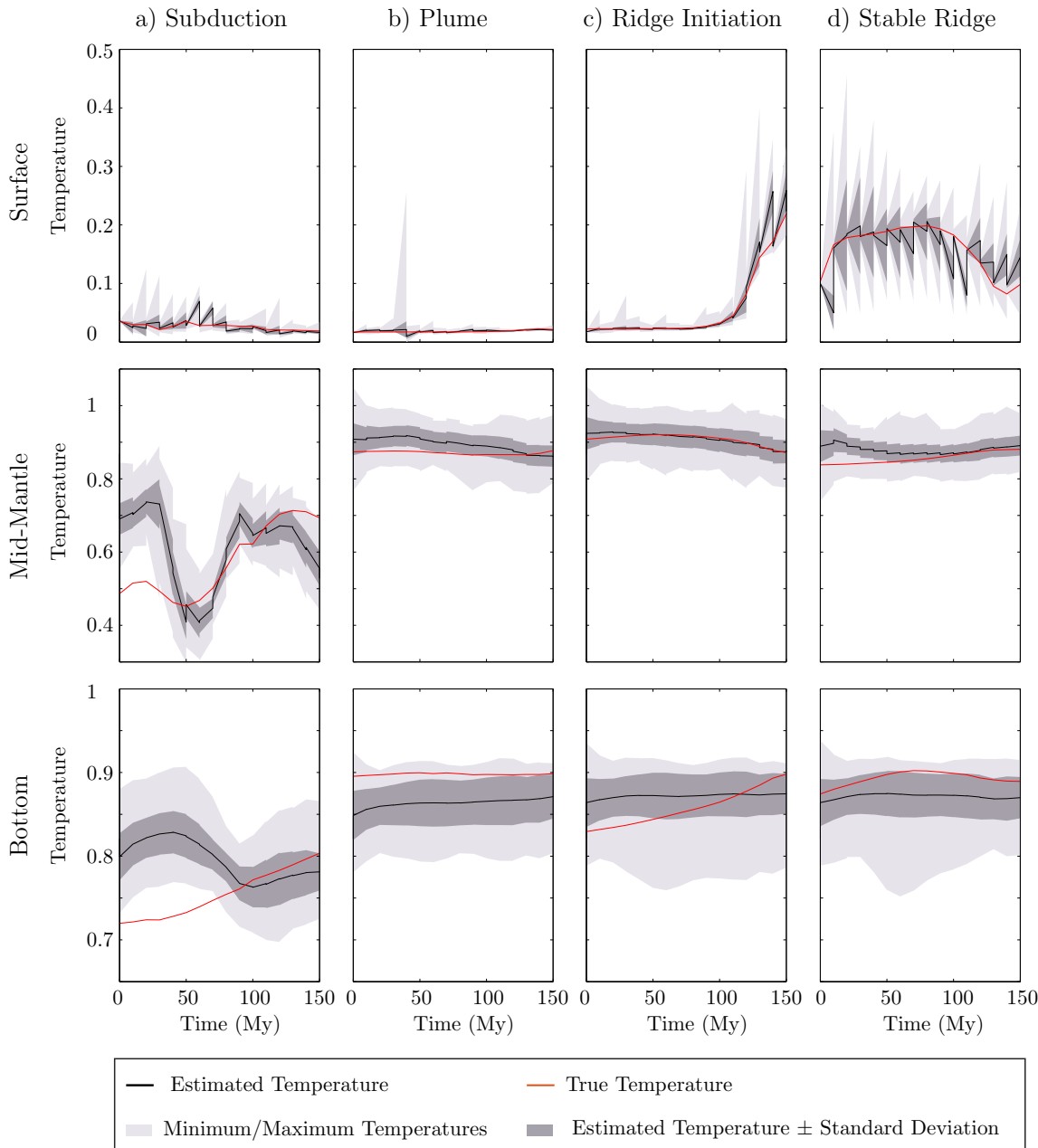

**Figure 11.** Detailed results of the assimilation depicted in Fig. 10, second row and Fig. 3(a). Each graph represents the time evolution of the temperature value at points on four profiles, corresponding to different geodynamic contexts: a) a subduction (first column), b) an upwelling (second column), c) a ridge initiation (third column), and d) a stable ridge (fourth column). The first row corresponds to points at the surface of the domain, the second row to points in the mid mantle and the third row to points at the bottom of the domain. The lateral coordinates of the points are shown in Fig. 10, second row first column. The red line is the true temperature, the black line is the average of the ensemble, the dark grey area represent the average plus or minus the standard deviation of the ensemble, and the light gray area is the area spanned by the minimum and the maximum value taken by the ensemble of 288 members.

tion, with current developments including the implementation of chemistry, nonlinear rheologies, elasticity, phase transition and compressibility (see e.g. Zhong et al., 2015, for a review of recent developments of mantle convection codes). In particular, this ease of implementation allows us to work on models producing self-consistently plate-like tectonics at their surface, and hence to obtain forecasts whose data can be ultimately compared with plate reconstructions.

The application of the ensemble Kalman filter to the mantle circulation problem is the continuation of the simpler sequential filter that we developed in an earlier work (Bocher et al., 2016). The main difference between the two filters is that the EnKF evaluates the state covariance matrix with an ensemble of members. This ensemble approach allows the nonlinear evolution of errors during the forecast stage. This leads to a higher precision in the reconstruction, but also to a more robust scheme, able to reconstruct evolutions which could not be reconstructed with the former method (as illustrated by Fig. 4 and 5). Moreover, the

ensemble assimilation provides an estimate of the uncertainty on the reconstruction at each point of the domain. The estimation of uncertainties could be a valuable information for plate tectonic reconstructions, especially for regions and times where data are scarce, because they show the different possible scenarios supported by the ensemble.

This gain in information and quality for reconstructions comes with a computational price. While we could perform the former assimilation method in one core hour, the method developed here requires several hundreds to several thousands of

core hours. However, an efficient parallelization using the PDAF software (Nerger and Hiller, 2013) in combination with the parallel code STAGYY produces a highly parallel ensemble filter, able to perform the assimilations on 768 cores in 20 min for an ensemble of 96 members and 3 hours for an ensemble of 768 members.

The important computational cost of the EnKF limited us in the number of assimilations we could test. After checking the stability of the assimilation results on four different evolutions, we chose to focus on studying the effect of the parameters of the

ensemble data assimilation: the size of the ensemble and the vertical and horizontal correlation lengths. We found that the best compromise between the accuracy of the assimilation and the computational cost was an ensemble of 288 members (among the ensemble sizes we tested, i.e. 96, 288 and 768). Indeed, between the assimilations with 288 and 768 members, the global average error on the temperature field (as defined in Eq. 43) decreases by 0.0013 while the size of the ensemble (and hence the computational cost) is multiplied by 2.7. On the contrary, dividing the size of the ensemble by 3 (from $N = 288$ to 96) leads

to an increase of the global average error of 0.0086. These differences in errors appear to be small, however they affect the quality of the reconstruction of thermal structures. We can see this in Fig. 10 for example: the global errors on temperature (as defined in Eq. 42) range between 0.0367 and 0.0461, so the difference in global errors is at most 0.0094. Locally, this translates into the presence (or absence) of artifactual geodynamic structures (like ridges and upwellings, visible in the second column of Fig. 10). Covariance localization proved to be important to minimize the error in the reconstruction of mantle structure: as

shown in Fig 9, for 288 members, the difference in the average error is of 0.0065 between the optimal correlation length and the least favorable one.

During these tests, we also evaluated how accurate the estimation of uncertainties (i.e. the spread of the ensemble) is with respect to the true error, and more generally, how reliable the forecast (i.e. the ensemble) is. If we consider the four assimilations with different data time series presented in Fig. 3, the true global error on temperature is higher than the ensemble spread in

three cases. This would indicate that we are on average overconfident in our forecasts. To test in more details the reliability of

the ensemble, we produced rank histograms for temperature at the surface, mid-depth and at the bottom of the domain (Fig. 7). The rank histogram corresponding to surface temperature does not detect any biases or over/underspread in the ensemble. On the contrary, the rank histograms are significantly nonuniform at depth. Our interpretation is that this tendency is linked to the configuration of the data assimilation problem, combined with the simple scheme used for covariance inflation (Sect. 3.2).

Indeed, the inflation factor which we propose is directly linked to the innovation statistics, and it is spatially uniform. It follows that the inflation factor will correct adequately the spread of the ensemble at the surface, where the data are located, but not necessarily at depth, where no observation is available. To improve the reliability of the ensemble at depth, a solution could be to implement a more complex algorithm for the inflation factor, especially spatially varying inflation as proposed by Anderson (2009) and Miyoshi (2011) for example.

Another important question for future applications with Earth data is: how well can we assess the quality of an assimilation when only observed data are available, i.e without any knowledge on the true state? To answer this question, we investigated the statistics of the cumulative innovation and of the instantaneous innovation for different ensemble sizes and correlation lengths. The variation of both cumulative innovation and instantaneous innovation as a function of ensemble size show the same tendency as the global average error on the temperature field: the larger the ensemble, the lower the instantaneous

and cumulative innovations, and results for $N = 288$ and 768 are very close, (see Fig. 2). On the contrary, the correlation lengths minimizing the norm of the cumulative innovation and the instantaneous innovation were different from the ones minimizing the error on the temperature field. This shows the limits of these indicators to determine the optimal parameters for the assimilation. In a realistic case, rigorous a posteriori evaluation of a data assimilation result would require comparison of the prediction made with independent observations (Talagrand, 2014). For mantle circulation, seismic tomography, topography,

true polar wander or the geoid could play this role.

By construction, sequential data assimilation methods do not propagate new information back in time. In the case of the reconstruction of mantle circulation, this is a clear disadvantage since the information on the Earth's surface tectonics tends to become more reliable as we get closer to present-day. Consequently, a natural extension of the present work would be to implement an ensemble Kalman smoother (Evensen and Van Leeuwen, 2000; Van Leeuwen, 2001). In the same way as the

EnKF uses sample spatial correlations of the ensemble to update the state of the system with new observations, the ensemble Kalman smoother uses sample time and space correlations of successive ensembles to update former states with the new observations. Evensen (2003) shows how an ensemble Kalman smoother can be implemented with a minimal computational cost alongside a preexisting EnKF. Moreover, Nerger et al. (2014) shows that such algorithm is efficient for nonlinear models, and that in their test case, optimal localization parameters for the ensemble Kalman smoother coincide with optimal localization

parameters for the EnKF.

As a first approach to test the EnKF for mantle circulation reconstructions, we chose a fairly simple convection model. As already discussed in Sect. 2.1, a more realistic mantle model would have, among other things, a 3D-spherical shell geometry and a higher Rayleigh number. This would substantially increase the size of the data assimilation problem. However, we followed the procedure as described in Nerger and Hiller (2013) to implement the EnKF. This results in a highly scalable filter,

enabling the computation of the EnKF assimilation in a reasonable time. An increase in the Rayleigh number also implies

thiner boundary layers, slabs and plumes. This could translate into lower optimum correlation lengths for the EnKF. A more realistic model would additionally include a viscosity increase in the lower mantle (Ricard et al., 1993), and the presence of continents. This would tend to lengthen the wavelength of convection in the lower mantle and therefore might ease the mantle circulation reconstruction (see for example Ricard, 2015, Sect. 7.02.6.3.2 and 7.02.6.7 for a discussion of both effects on
mantle convection).

In the synthetic experiments of Sect. 4, the convection model used to produce the series of data is the same as the forward model used during the assimilation. For an application with Earth data, this will not be the case. The equations solved in models of mantle convection still hold some shortcomings (Ricard, 2015). Moreover, theories, observations and experiments do not yet fully constrain parameters, especially rheological ones (King, 2016), and variations in rheology affect the reconstructions of
mantle circulation (Bello et al., 2015). Hence it could be fundamental to take into account model errors. A first order solution is to increase the inflation parameter $\gamma$ in Eq. (32): this would overall increase the a priori uncertainty on the mantle estimation. Performing experiments where the model used to compute the observation is different from the model used for the assimilation would provide us with more information on how to implement model errors. Another solution would be to consider the joint assimilation of the state and model parameters. Although it is in principle possible for the EnKF (Evensen, 2009b), it could
be computationally not tractable. Indeed, the response of mantle dynamics to different rheological parametrization is highly nonlinear, and their inversion calls for the development of techniques focusing on rheology, such as adjoint based inversions of rheological parameters (Worthen et al., 2014; Ratnaswamy et al., 2015) or further applications of the recently developed pattern recognition techniques for mantle convection (Atkins et al., 2016).

The choice of the synthetic experiments assimilation window of 150 Myr is a compromise between having the possibility
to compute assimilations for various cases and having an assimilation window covering most part of the timespan of plate tectonic reconstructions (Seton et al., 2012; Müller et al., 2016; Torsvik et al., 2010). However, the structure of the dataset used for the synthetic experiments is a very idealized version of the actual plate reconstruction models. We already discuss this issue in Bocher et al. (2016). In the following, we supplement and update this discussion in the light of research that has recently come to the fore.

First, we set a time series of data covering the whole surface of the domain and regularly available, every 10 Myr. Plate tectonic reconstructions data are more complex. They are based on the estimation of finite relative rotations between individual plates, structured into a hierarchy describing global relative motions and anchored in an absolute reference frame. The span of each finite relative rotation is determined depending on the amount and quality of information available for a specific context and therefore varies depending on plate pairs and times. The average span of finite rotations of recent plate models
is of the order of 10 Myr (Torsvik et al., 2010) to 5 Myr (Müller et al., 2016), but varies over time with for example 1 Myr resolution for the last 20 Myr in some regions (Merkouriev and DeMets, 2014), or some gaps in the data such as during the cretaceous superchron from 121 to 83 Myr ago (Granot et al., 2012). The continuously closed plate algorithm (Gurnis et al., 2012) produces plate tectonic reconstruction maps continuous in space and time which allows the creation of a series of global plate reconstructions at regular intervals. Nonetheless, creating such a regularized time series of reconstruction might miss
tectonic events. Instead, we could adapt the frequency of analyses to the varying plate reconstruction resolution. Additional

synthetic experiments with a time-series of data whose frequency evolves through time are necessary to explore the limits of such method.

Second, the observations were perturbed independently with a Gaussian noise of $10\%$ of the respective root mean square value of surface heat flux and surface velocities. The estimation of uncertainties on absolute plate motion models involves estimation of both uncertainties in relative plate motion and on the absolute reference frame (Müller and Wessel, 2015). The main source of information on the motion of plates comes from the map of seafloor magnetic anomalies. Hellinger (1981) developed a method to compute relative motion of plates and associated uncertainties inferred from magnetic anomaly identifications. Recently, Seton et al. (2014) built an open source community database. It gathers seafloor magnetic anomaly identifications, and estimation with Hellinger (1981) method of plates relative motion and associated uncertainties. This database could be used in the future as a basis to automatically produce global plate motion histories and assess their uncertainties. To our knowledge, this has not been done so far at a global scale. On a regional scale and for recent time (5 to 20 Myr), Iaffaldano et al. (2012) applied the trans-dimensional hierarchical Bayesian method to reduce noise in finite rotation data and produce time series of high resolution plate relative motions. More recently, Iaffaldano and Bunge (2015) applied this technique to the relative motion of the pacific plate with North America for the last 75 Myr. The uncertainties on relative plates velocities ranges from 5 to $40\%$ of the root mean square surface velocity. As we go further back in time, the quantification of relative plate motion uncertainties becomes hazardous: most of the seafloor created before 150 Myr has been destroyed by subduction. These plate tectonic reconstructions involve interpretation of different types of data, with a limited spatial coverage and relies heavily on human expertise. For these epochs, maintaining very high uncertainties on the regions where few data supports the reconstructions would be a solution.

## 6 Conclusions

We applied the ensemble Kalman filter algorithm to the reconstruction of mantle circulation through time. We chose a formulation with covariance inflation and localization to minimize the effect of sampling errors in the estimation of the forecast error covariance matrix. Synthetic "twin" experiments with different evolutions and for different parameters allowed us to assess the efficiency of the algorithm and to determine the optimal parameters for the assimilation.

This work builds on the developments of a first approach to sequential data assimilation for mantle circulation made in Bocher et al. (2016). The EnKF is more robust and on average more accurate than the former method. Additionally, the ensemble Kalman filter provides not only an estimate of mantle circulation, but also detailed maps of uncertainties on this estimation.

We evaluate the accuracy of the EnKF as a function of three main parameters: the size of the ensemble, and two covariance localization parameters, namely the vertical correlation length and horizontal correlation angle. We find that a size of the ensemble of the order of 300 members is sufficient to have an accurate estimation of the evolution of the state. For this ensemble size, the optimal vertical correlation length corresponds to two thirds of the domain thickness, and the optimal

horizontal correlation angle is of $\pi/10$ (around 2000 km). These values should be reevaluated as the dynamical model becomes more realistic.

The EnKF was implemented using the parallel data assimilation framework PDAF in a preexisting mantle convection code, STAGYY. The resulting code is highly scalable, which means that the application of the EnKF to realistic data assimilation with plate reconstructions and a 3D spherical mantle model is within reach in a foreseeable future.

*Acknowledgements.* We thank the two anonymous reviewers as well as the editor, Olivier Talagrand, for their very useful comments and suggestions, which helped improve this manuscript. The research leading to these results has received funding from the European Research Council within the framework of the SP2-Ideas Program ERC-2013-CoG, under ERC grant agreement 617588. Calculations were performed using HPC resources from GENCI-IDRIS (grant 2016-047243). The contribution of Alexandre Fournier is IPGP contribution number XXX.

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

**Table 1.** Values of the parameters of the forward model

| Symbol | Meaning | value |
|---|---|---|
| $\text{Ra}_T$ | Rayleigh number based on temperature difference | $10^6$ |
| $\text{Ra}_H$ | Rayleigh number based on internal heating | $2.05\ 10^7$ |
| $L$ | number of grid points in longitude | 384 |
| $M$ | number of grid points in radius | 48 |
| $r_a$ | Radius of the top of the domain | 2.2 |
| $r_b$ | Radius of the bottom of the domain | 1.2 |
| $T_a$ | Temperature at the top of the domain | 0 |
| $T_b$ | Temperature at the bottom of the domain | 0.9 |
| $E_A$ | Activation Energy | 23.03 |
| $T_1$ | Temperature at which $\mu_T = 1$ | 1 |
| $\beta$ | Factor of viscosity reduction for partial melting | 10 |
| $T_{s_0}$ | Solidus Temperature at $r = r_a$ | 0.6 |
| $\nabla_r T_s$ | Radial gradient of the solidus temperature | 2 |
| $\sigma_Y$ | Yield Stress | $10^4$ |
| $\nabla_r \sigma_Y$ | Radial gradient of the yield stress | $2\ 10^5$ |

**Table 2.** Notations and dimensions of data assimilation variables

| Symbol | Meaning | Size (Literal) | Size (Value) |
|---|---|---|---|
| $\boldsymbol{x}$ | state | $LM + L$ | 18 816 |
| $\boldsymbol{y}$ | data | $L + L$ | 768 |
| $\mathbf{H}$ | observation matrix operator | $(L+L) \times (LM+L)$ | $768 \times 18\,816$ |
| $\mathbf{R}$ | observation error covariance matrix | $(L+L) \times (L+L)$ | $768 \times 768$ |
| $\mathbf{P}$ | state error covariance matrix | $(LM+L) \times (LM+L)$ | $18\,816 \times 18\,816$ |
| $\mathbf{X}$ | ensemble state | $(LM+L) \times N$ | $18\,816 \times N$, |
| | | | $(N = 96{,}288 \text{ or } 768)$ |

**Table 3.** Notations and range of values tested for data assimilation parameters

| Symbol | Meaning | value |
|---|---|---:|
| $N$ | number of ensemble members | 96 to 768 |
| $K$ | number of observation times | 16 |
| $\gamma^+$ | maximum inflation factor | 1.25 |
| $\ell_v$ | vertical correlation length | 0.3 to 1 |
| $\ell_h$ | horizontal correlation angle | $\pi/10$ to $\pi/2$ |