# Peer review of "Ensemble Kalman filter for the reconstruction of the Earth's mantle circulation"

_Nonlinear Processes in Geophysics, 2017_

## Referee Comment (RC1) · Anonymous Referee #1 · 12 Mar 2017

Summary This manuscript presents an observing system simulation experiment using an ensemble Kalman filter for an idealized model similar to those used to simulate mantle convection. Apparently this is one of the first applications of an ensemble filter in this field. The results suggest that there is potential for applying ensemble filters to mantle convection models. The novel issue is the behavior of this particular model and its relevance for more realistic models that could use real observations. Given this, more information about the error growth characteristics of this model would be a useful addition, as would additional references to more mature ensemble filter explorations in other areas of geoscience.

Major comments:

1. It's fine to introduce an established method to a new field, but there should be a

number of additional references to literature in other areas of geophysics where all of the tuning, methodology, and evaluation techniques have been more carefully documented.

2. The method used to create the initial ensemble is unclear and not common in the ensemble filtering literature for geosciences. The discussion of 'continuous' versus sample P matrices seems to be part of the confusion. Normally, one wouldn't really know P outside of the ensemble filter context, except perhaps in a loose 'climatological' sense. General discussion of P's versus ensemble sample P's is fuzzy. The discussion starting at the end of p. 7 seemed particularly confusing. Apparently there are 400 'climatological' samples. These will only span a phase subspace of at most 399 dimensions. However, an eigenvector analysis somehow produces 1928 distinct eigenvalues. An SVD analysis, the common way to filter a sample covariance in general, would give 399 or fewer. After the eigenanalysis is completed, an initial analysis step is somehow done (continuous, not ensemble?) and finally the analysis covariance is somehow sampled to generate an initial ensemble. Additional clarity is needed here. Also, doing anything more than filtering the sample covariance from the 400-member sample seems inappropriate. Can you relate this to a similar procedure in the literature from a more mature ensemble field?

3. The Kalman Filter and ensemble variants basically depend on exponentially growing directions in the model phase space to work effectively. For novel applications, it is important to know something about the growth of error. Many geophysical applications will include an experiment where ensembles are evolved without assimilation to demonstrate that there is ensemble error growth and to show that the assimilation improves on this control case. I suggest including results from such a case. Easiest way would be just to use one or more of the existing ensemble initial conditions, but it can also be done with smaller ensembles to just explore error growth.

4. Although it happens far too often in the literature, looking at analysis innovations is simply bad practice. It is almost impossible to interpret the results as you demonstrate with your figure 7. There is, however, a simple alternative that is much easier to interpret: look at forecast innovations. The forecast observations are independent of the forecast and so should give results consistent with comparisons to truth (although noisier of course). You can, of course, also easily simulate withheld (not assimilated) observations and compare your analysis to these. However, given the dearth of available observations it is unlikely that this is what you would choose to do in a real data experiment.

5. I suspect that you will find that a deterministic ensemble filter will produce significantly better results for your problem with small (say less than 100) ensemble sizes. The additional sampling error may be a primary cause of the 96 member ensemble being significantly worse than the large ones.

6. Repeated claims are made that the 288-member ensemble (or an ensemble of size about 300) is optimal/optimum. These claims are unsupported. A norm is not established and it is clear that you are doing some intuitive combination of quality and cost. In addition, having only 3 ensemble sizes gives you no basis for claiming that the middle one is optimal. You would need to try additional cases.

7. The adaptive inflation approach you are using is fairly simplistic. More robust methods that do an evolving Bayesian estimation for inflation have been described in papers like Miyoshi, T., 2011: The Gaussian Approach to Adaptive Covariance Inflation and Its Implementation with the Local Ensemble Transform Kalman Filter. Mon. Wea. Rev., 139, 1519-1535, and ANDERSON, J. L. (2009), Spatially and temporally varying adaptive covariance inflation for ensemble filters. Tellus A, 61: 72–83. doi:10.1111/j.1600-0870.2008.00361. It is clear that almost all of your ensembles are significantly under dispersed and improved performance should result from better inflation. This may be particularly important in improving the 96-member case, too.

8. Doing all interpretations with normalized error and spread is not common and can make interpretation of relative capabilities complicated. Certainly, the discussion at the
end of the paper referring to percentage errors is confusing and potentially misleading. I suggest at least including a few results that look at the unnormalized RMSE, etc.

9. You briefly look at the range of the ensemble and whether it bounds the truth (end of p. 22). This type of evaluation is misleading since how often the truth should be bounded is a function of the ensemble size. A rank histogram analysis would be more appropriate and would also have the potential to reveal more about challenges being faced by the ensemble assimilation.

Minor points:

1. p. 3, line 9: The reference to Evensen 1994 should include a caveat that it is not a correct derivation of the EnKF and that the 1998 Burgers et al presents the correct algorithm.

2. P. 6, line 8: "data that ARE"

3. P. 7, equation 11: Using an extended (or joint) state is okay. However, you don't really motivate why. The most standard practice would have been to include all the observation priors in the joint state. Include a brief discussion of why you made this choice.

4. P. 7, line 12 and other places: "expectancy" To the best of my knowledge, this word is not being used correctly here. "expected value" might be better.

5. P. 7, line 12: This sentence is confusing since you never really compute the P's, why even refer to them? They are unknown and unknowable in some sense.

6. P. 7, line 19: Note that the velocity forward operator is just a vector extraction (identity).

7. P. 8, line 20: What does 'efficient' mean here. Should become irrelevant anyway if this is behaving like a KF/EnKF; the initial ensemble choice should lose any qualitative impact as the filter proceeds. If this is not the case, then a Kalman filter class algorithm

may not be a particularly good choice.

8. P. 10, line 12: This statement is too strong. The Janjic approach can be significantly more efficient in some parallel computation situations. In the simple PDAF implementation, this may not be the case.

9. P. 10, sentence starting on line 25: I cannot understand this sentence. Not sure what "direct forecast error localization" means here.

10. P. 11, line 9: Replace "noise" with "add noise to"

11. P. 11, line 10: What is the root mean square of surface heat flux... Instantaneous, variation over model?

12. Figure 1 caption: "150 Myr OBSERVATION dataset" makes it clear that you are just using the synthetic observations.

13. Figure 1: Note that the error seems to be going back up towards the end of the time series. This probably merits a comment in the text. I suspect it is due to the insufficient spread.

14. P. 12, line 6: Not sure what is meant by a "stabilization"

15. Start of p. 13, discussion of parallel efficiency. This discussion is inappropriate without lots more detail about the computing resources used. A good parallel implementation of an EnKF should scale very well (embarrassingly parallel) for a problem like this, so I was surprised that the time wasn't very nearly constant with a sufficient number of cores.

16. P. 13, line 13: rms values over what sample?

17. P. 13, line 16: Not sure what "at first order" means here.

18. P. 13, line 20: An estimate of the uncertainty, not the error, would be more common usage to describe the spread.

19. Figure 2: I don't see how these can be consistent with figure 1 which shows N=96 much worse by the 10th assimilation time.

20. Figure 3: These are dangerously under dispersed in 3 cases.

21. P. 17, line 2: It's the ensemble covariance that matters, not the scales of spatial variability. These may or may not be the closely related.

22. P. 19, line 13: Are the figure 8 results for the best localizations?

23. Figure 7 caption: What is 'K'?

24. Figure 9 caption: Need more caption info. What assimilation? Max and min temps from ensemble members? How many of them?

25. P. 25, line 13: This looks like very poor parallel behavior to me, unless this is somehow dominated by the model forecast times.

26. P.26, line 7: This is too simplistic. In a smoother for a problem with things that are advecting/convecting, an observation at the current time will have largest correlation with a point upstream at an earlier time. The localization needs to be shifted away from the observation as a function of time lag and the maximum value should be less than 1.

27. P. 26, line 27: Why do you think this? Are there more parameters than state variables?

28. P. 26, line 34: You think you are not converged? Plots look like you've bottomed out and error is increasing as a function of assimilation time.

---

## Referee Comment (RC2) · Anonymous Referee #2 · 25 Aug 2017

This paper on the testing of an Ensemble Kalman Filter (EnKF) for mantle convection circulation, using twin experiments is well written and relatively clear. The results are valuable and will eventually lead to more precise assimilation data of the Earth's mantle into mantle convection models. I am suggesting relatively minor corrections (no further simulations), but rather some improvements for clarity and presentation.

The comments are:

1. The discussion of the error plots was at times confusing. In some places you simply refer to errors, when you could mean the difference between the true state and the assimilation, or sometimes the innovation (equation 36 for example, which isn't really a forecast error). Please be clear about what error you mean each time you use this term.

2. I was surprised by how little the errors dropped in Figure 1, until later in the results and discussion it became apparent that only small regions have most of the errors (like the plumes, ridge or subduction). It would be really helpful to plot the average error over these regions rather than the entire domain (where the temperature field is fairly constant for long periods). I think this would give a clearer picture of the errors between the various experiments.

3. It would also be really useful to see how the velocity field responds to the assimilation, because this is the part of the state directly related to the surface velocity. I realize that it is not a prognostic variable, but it is an important part of the state.

4. Please define the vector {\bf 1} in equation 18.

5. The text is pretty carefully edited for writing and typos. I just found a couple of things: line 12 , change explicitely to explicitly (though this suggests that you didn't run a spell check, so there might be more). And page 13, line 3, the word "embarassingly" is probably not appropriate.

6. Please clarify what you mean by state space localization, page 10, line 16.

7. Some of the figures need larger fonts on the captions, particularly Figure 6. And if possible, use the Greek symbol for pi.

---

## Author Comment (AC2) · 31 Oct 2017

We thank the anonymous reviewer for his/her review of our paper. We hope that the answers provided here, as well as the modifications proposed in the paper will be satisfactory. Please find below the list of comments, each associated with our answer and details on the associated modifications of the manuscript

1. The discussion of the error plots was at times confusing. In some places you simply refer to errors, when you could mean the difference between the true state and the assimilation, or sometimes the innovation (equation 36 for example, which isn't really a forecast error). Please be clear about what error you mean each time you use this term.

[Figure]

⇒ **We changed the notations for innovations to a system which is hopefully more straightforward. We also changed the name of variables based on the innovation vector that were refered to as "errors".**

2. I was surprised by how little the errors dropped in Figure 1, until later in the results and discussion it became apparent that only small regions have most of the errors (like the plumes, ridge or subduction). It would be really helpful to plot the average error over these regions rather than the entire domain (where the temperature field is fairly constant for long periods). I think this would give a clearer picture of the errors between the various experiments.
⇒ **We found a compromise between this suggestion and the major comment number 8 of reviewer 1. We changed all the plots to represent the RMS error and plot on each figure the RMS error that we would obtain if the estimate was the "climatological" average 1D profile.**

3. It would also be really useful to see how the velocity field responds to the assimilation, because this is the part of the state directly related to the surface velocity. I realize that it is not a prognostic variable, but it is an important part of the state.
⇒**Overall, the surface velocities are very well corrected during analyses, due to their direct link with observations. We plotted on figure 1 the evolution of errors on Velocities.**

4. Please define the vector **1** in equation 18.
⇒ **The text has been modified.**

5. The text is pretty carefully edited for writing and typos. I just found a couple of things: line 12 , change explicitely to explicitly (though this suggests that you didn't run a spell check, so there might be more). And page 13, line 3, the word "embarassingly" is probably not appropriate.
⇒ **The text has been modified.**

6. Please clarify what you mean by state space localization, page 10, line 16.
⇒**We mean that the localization has to be done on the forecast error covariance matrix. Text modified.**

7. Some of the figures need larger fonts on the captions, particularly Figure 6. And if possible, use the Greek symbol for pi.
⇒**The text has been modified.**
* * *

---

## Author Response (AR1)

**Answer to the comment of anonymous referee #1**

We thank the anonymous reviewer for his/her very complete review of our paper. We really appreciate the time investment that it must have been, and hope that the answers provided here, as well as the modifications proposed in the paper will be satisfactory. Please find below the list of comments, each associated with our answer and details on the associated modifications of the manuscript.

**1 Major Comments**

1. It's fine to introduce an established method to a new field, but there should be a number of additional references to literature in other areas of geophysics where all of the tuning, methodology, and evaluation techniques have been more carefully documented.

 $\Rightarrow$ We added reference to the litterature when deemed necessary (10 new data assimilation references have been added). If you feel that a specific reference is missing, please let us know.

- 2. The method used to create the initial ensemble is unclear and not common in the ensemble filtering literature for geosciences. The discussion of 'continuous' versus sample P matrices seems to be part of the confusion. Normally, one wouldn't really know P outside of the ensemble filter context, except perhaps in a loose 'climatological' sense. General discussion of P's versus ensemble sample P's is fuzzy. The discussion starting at the end of p. 7 seemed particularly confusing. Apparently there are 400 'climatological' samples. These will only span a phase subspace of at most 399 dimensions. However, an eigenvector analysis somehow produces 1928 distinct eigenvalues. An SVD analysis, the common way to filter a sample covariance in general, would give 399 or fewer. After the eigenanalysis is completed, an initial analysis step is somehow done (continuous, not ensemble?) and finally the analysis covariance is somehow sampled to generate an initial ensemble. Additional clarity is needed here. Also, doing anything more than filtering the sample covariance from the 400-member sample seems inappropriate. Can you relate this to a similar procedure in the literature from a more mature ensemble field?  $\Rightarrow$  We are not simply computing the sample correlation matrix, but use the symmetries of the problem to better estimate the covariance matrix. That is why we do not do an SVD but an EVD instead and obtain more than 399 eigenvalues. We rephrased the paragraph to make this point clear. We also cite the paper where the procedure is explained in more details (Bocher et al., 2016).
- 3. The Kalman Filter and ensemble variants basically depend on exponentially

growing directions in the model phase space to work effectively. For novel applications, it is important to know something about the growth of error. Many geophysical applications will include an experiment where ensembles are evolved without assimilation to demonstrate that there is ensemble error growth and to show that the assimilation improves on this control case. I suggest including results from such a case. Easiest way would be just to use one or more of the existing ensemble initial conditions, but it can also be done with smaller ensembles to just explore error growth.

 $\Rightarrow$  This work has already been done and discussed extensively in Bello et al. (2014). We reworked section 2.1 on the mantle convection model, adding more explanations on the choice of the model, and adding a paragraph on the chaotic nature of mantle convection, the thermal turbulence that affects it, and the result of twin experiments measuring error growth for our model.

4. Although it happens far too often in the literature, looking at analysis innovations is simply bad practice. It is almost impossible to interpret the results as you demonstrate with your figure 7. There is, however, a simple alternative that is much easier to interpret: look at forecast innovations. The forecast observations are independent of the forecast and so should give results consistent with comparisons to truth (although noisier of course). You can, of course, also easily simulate withheld (not assimilated) observations and compare your analysis to these. However, given the dearth of available observations it is unlikely that this is what you would choose to do in a real data experiment.

 $\Rightarrow$  We are not looking at the analyzed innovation, but forecast innovation, as defined in equation 21 of the original manuscript, or equation 44 of the revised version. The x-axis title of figure 2 "number of analyses" might have been confusing, so we changed it to "forecast number".

5. I suspect that you will find that a deterministic ensemble filter will produce significantly better results for your problem with small (say less than 100) ensemble sizes. The additional sampling error may be a primary cause of the 96 member ensemble being significantly worse than the large ones.

 $\Rightarrow$  We actually started implementing a deterministic ensemble filter. However, as is said on page 10, second paragraph of the submitted manuscript, and validated on figure 6 for example, we need to apply localization on both the horizontal and vertical direction. Since the observations are only located at the surface, we need to apply localization in the state space. To do so with a deterministic filter would require choices on the resampling after analysis that we could not justify properly, which is why we did not implement and test it yet.

6. Repeated claims are made that the 288-member ensemble (or an ensemble of size about 300) is optimal/optimum. These claims are unsupported. A norm is not established and it is clear that you are doing some intuitive combination of quality and cost. In addition, having only 3 ensemble sizes gives you no basis for claiming that the middle one is optimal. You would need to try additional cases.

 $\Rightarrow$  We agree and rephrased the two occurences were we used the word optimal for the ensemble size.

7. The adaptive inflation approach you are using is fairly simplistic. More robust methods that do an evolving Bayesian estimation for inflation have been described in papers like Miyoshi, T., 2011: The Gaussian Approach to Adaptive Covariance Inflation and Its Implementation with the Local Ensemble Transform Kalman Filter. Mon. Wea. Rev., 139, 1519-1535, and ANDERSON, J. L. (2009), Spatially and temporally varying adaptive covariance inflation for ensemble filters. Tellus A, 61: 72–83. doi:10.1111/j.1600-0870.2008.00361. It is clear that almost all of your ensembles are significantly under dispersed and improved performance should result from better inflation. This may be particularly important in improving the 96-member case, too.

 $\Rightarrow$  The adaptive inflation, although simplistic, corrects rather successfully the forecast variance of the temperature as soon as it is close to observations (see rank histogram of figure 7a of the revised manuscript: we have a slight bias of the ensemble towards colder temperatures, but the ensemble is not under-dispersed at the surface). However, in depth, the ensemble is under-dispersed and our interpretation is that, since observations are only at the surface, we do not correct the ensemble spread adequately in depth, and any adaptive inflation scheme based on the innovation statistics will not improve the spread of the ensemble in depth. We added a paragraph on the possible improvement of adaptive inflation in the discussion, and added a subsection on rank histograms in the section 4:A posteriori evaluation of the ensemble Kalman filter method.

8. Doing all interpretations with normalized error and spread is not common and can make interpretation of relative capabilities complicated. Certainly, the discussion at the end of the paper referring to percentage errors is confusing and potentially misleading. I suggest at least including a few results that look at the unnormalized RMSE, etc.

 $\Rightarrow$  Our aim was to provide the reader with a reference point, since the temperature is non dimensional in our models. We agree that this formulation made the whole discussion on results confusing, so we changed all the figures to plot RMSE instead of the RMS of the normalised error. To provide a point of reference for the error, we plotted also on the figures 1 and 3 the error that we would have made if we had supposed a 1D temperature profile corresponding to the average temperature field computed from a very long free run.

9. You briefly look at the range of the ensemble and whether it bounds the truth (end of p. 22). This type of evaluation is misleading since how often the truth should be bounded is a function of the ensemble size. A rank histogram analysis would be more appropriate and would also have the potential to reveal more about challenges being faced by the ensemble assimilation.
⇒ Indeed, this was a mistake on our part. We deleted sentences referring to this in the text. We performed a rank histogram analysis for the temperature, heat flux and velocities, the results are described in section 4.3 and shown in figures 6 and 7 of the revised manuscript.

**2 Minor Comments**

- p. 3, line 9: The reference to Evensen 1994 should include a caveat that it is not a correct derivation of the EnKF and that the 1998 Burgers et al presents the correct algorithm.
   ⇒Text has been modified
- P. 6, line 8: "data that ARE" ⇒Text has been modified
- 3. P. 7, equation 11: Using an extended (or joint) state is okay. However, you don't really motivate why. The most standard practice would have been to include all the observation priors in the joint state. Include a brief discussion of why you made this choice.

 $\Rightarrow$  We rephrased the paragraph in question to clarify the use of the augmented state to avoid a nonlinear observation operator, added a reference to Evensen 2003, and a reference to the paragraph where the observation operator is discussed.

- 4. P. 7, line 12 and other places: "expectancy" To the best of my knowledge, this word is not being used correctly here. "expected value" might be better.
  ⇒Text has been modified
- 5. P. 7, line 12: This sentence is confusing since you never really compute the P's, why even refer to them? They are unknown and unknowable in some sense.

 $\Rightarrow$  We corrected the sentence, accounting for the fact that we actually compute the covariance matrices, but just for the initialization step. As discussed in major comment 2,  $P_1^f$  is the background covariance matrix, computed from a free run (the "climatology"), so we know it, if we consider the model to be perfect.

- 6. P. 7, line 19: Note that the velocity forward operator is just a vector extraction (identity).
  - $\Rightarrow$ Text has been modified

rephrased the sentence accordingly.

- 7. P. 8, line 20: What does 'efficient' mean here. Should become irrelevant anyway if this is behaving like a KF/EnKF; the initial ensemble choice should lose any qualitative impact as the filter proceeds. If this is not the case, then a Kalman filter class algorithm may not be a particularly good choice.
  ⇒ Efficient in the sense that the errors are smaller at the beginning of the assimilation and decrease faster. This is important for our problem, since the spin up time of the assimilation is of the same
- 8. P. 10, line 12: This statement is too strong. The Janjic approach can be significantly more efficient in some parallel computation situations. In the

order as the total timespan for which we have observations. We

simple PDAF implementation, this may not be the case.  $\Rightarrow$ **Text has been modified**

- 9. P. 10, sentence starting on line 25: I cannot understand this sentence. Not sure what "direct forecast error localization" means here.
  ⇒ we meant we apply localization directly on the forecast error covariance matrix, as opposed to the domain localization already implemented in PDAF, text has been modified accordingly.
- 10. P. 11, line 9: Replace "noise" with "add noise to"  $\Rightarrow$ **Text has been modified**
- 11. P. 11, line 10: What is the root mean square of surface heat flux... Instantaneous, variation over model?
  ⇒ the root mean square of surface heat flux and velocity are long term averages computed from the results of a free run. They are characteristic of the dynamics of the system. Text modified with these precisions.
- 12. Figure 1 caption: "150 Myr OBSERVATION dataset" makes it clear that you are just using the synthetic observations.
   ⇒Text has been modified
- 13. Figure 1: Note that the error seems to be going back up towards the end of the time series. This probably merits a comment in the text. I suspect it is due to the insufficient spread.

 $\Rightarrow$  The text has been modified to acknowledge the error growth at the end of the time series. Additionally, we show now in Figure 1 the evolution of the error of the surface velocity, as suggested by reviewer # 2. It shows that, contrary to the temperature, the error on velocity does not grow at the end of the time series. We also discuss in more details the reliability of the ensemble forecast in section 4.3 of the revised manuscript, using rank histograms.

- 14. P. 12, line 6: Not sure what is meant by a "stabilization"  $\Rightarrow$  We deleted the word stabilization and changed the description of figure 1: we identify 2 phases: rapid decrease of error and then slow growth of error (which we described as stabilization in the former version)
- 15. Start of p. 13, discussion of parallel efficiency. This discussion is inappropriate without lots more detail about the computing resources used. A good parallel implementation of an EnKF should scale very well (embarrassingly parallel) for a problem like this, so I was surprised that the time wasn't very nearly constant with a sufficient number of cores.

 $\Rightarrow$  The time is indeed nearly constant provided we have a sufficient number of cores, we meant here CPU time and not real elapsed time. We rephrased the sentence to make it clear that we evaluate the quality of the data assimilation against its computational cost.

16. P. 13, line 13: rms values over what sample?  $\Rightarrow$  We added precisions in the text (see also minor comment 11) 17. P. 13, line 16: Not sure what "at first order" means here.

 $\Rightarrow$  We meant that the cumulative mean innovation check is not a comprehensive test, but allows only a partial check of consistency. We rephrased the sentence.

- 18. P. 13, line 20: An estimate of the uncertainty, not the error, would be more common usage to describe the spread.
  ⇒ Text has been modified
- 19. Figure 2: I don't see how these can be consistent with figure 1 which shows N=96 much worse by the 10th assimilation time.  $\Rightarrow$  Figure 1 represents the error on the whole temperature field. Figure 2 shows statistics on the innovation, so the difference between observed and forecast surface velocities and heat fluxes. This means that although the forecast and the observed data at the surface are close, the estimated temperature field at depth differs from the true temperature field much more for N=96 than for N=288 or 768. We added this remark to the paragraph commenting on figure 2. We also reorganized the whole description of figure 2, for more clarity.
- 20. Figure 3: These are dangerously under dispersed in 3 cases.
   ⇒ We added a remark in the result section, study more precisely where the ensemble is biased/underdispersed with rank histograms, and rediscuss underdispersion in the discussion.
- 21. P. 17, line 2: It's the ensemble covariance that matters, not the scales of spatial variability. These may or may not be the closely related.  $\Rightarrow$  We agree. However, the ensemble covariance will be affected by the way small perturbations evolve and grow in the system. In our system, a slight temperature perturbation in the upper boundary layer can lead to the development of a new plate boundary. This links our discussion of spatial variability due to the structure of plate boundaries to the ensemble covariance matrix. Text has been complemented to make this link clearer.
- 22. P. 19, line 13: Are the figure 8 results for the best localizations?  $\Rightarrow$  Yes, we added this precision in the legend of Figure 8.
- 23. Figure 7 caption: What is 'K'?  $\Rightarrow$  K=16, legend updated.
- 24. Figure 9 caption: Need more caption info. What assimilation? Max and min temps from ensemble members? How many of them?
   ⇒Text has been modified
- 25. P. 25, line 13: This looks like very poor parallel behavior to me, unless this is somehow dominated by the model forecast times.
  ⇒ We already stated that, indeed, "during the assimilation of a dataset, most of the computational time is dedicated to the forecast step".

26. P.26, line 7: This is too simplistic. In a smoother for a problem with things that are advecting/convecting, an observation at the current time will have largest correlation with a point upstream at an earlier time. The localization needs to be shifted away from the observation as a function of time lag and the maximum value should be less than 1.

 $\Rightarrow$  Yes, we agree that a potential smoother could benefit from shifting the localization away from the observation. However, we might already gain some information by applying a simple localization for the smoother. Nerger et al. (2014) obtain encouraging results with this type of localisation in a large-scale ocean circulation model for example.

27. P. 26, line 27: Why do you think this? Are there more parameters than state variables?

 $\Rightarrow$ Not necessarily, but, as explained in the following sentence, the relationship of mantle dynamics to different rheological parameters is highly nonlinear: most likely, we will need very large ensembles to determine accurately the parameters.

28. P. 26, line 34: You think you are not converged? Plots look like you've bottomed out and error is increasing as a function of assimilation time.  $\Rightarrow$ We agree, we deleted the sentence.

[revised manuscript text omitted]

$$\boldsymbol{\sigma} = 2\mu_{\text{eff}} \dot{\boldsymbol{\epsilon}} = \mu_{\text{eff}} \left( \nabla \boldsymbol{u} + \left( \nabla \boldsymbol{u} \right)^T \right). \tag{4}$$

The choice of the effective viscosity  $\mu_{\text{eff}}$  takes into account both a viscous Newtonian behavior with a viscosity is crucial for the development of plate-like tectonics at the surface of the convective system. We choose for  $\mu_{\text{eff}}$  a composite rheology with a

25 viscous Newtonian component  $\mu_n$  and a pseudo-plastic behavior component, implemented with an equivalent "pseudo-plastic viscosity"  $\mu_y$ , such that

$$\mu_{\text{eff}} = \min(\mu_n, \mu_y). \tag{5}$$

The Newtonian viscosity  $\mu_n$  follows an Arrhenius law

$$\mu_n = \mu_0 \exp\left(\frac{E_A}{T + T_1}\right) \tag{6}$$

with  $\mu_0 = \exp\left(-\frac{E_A}{2T_1}\right)$ ,  $T_1$  the temperature at which the nondimensional  $\mu_n = 1$ , and  $E_A$  the nondimensional activation energy. We This law reflects the thermal activation of crystal deformation, and creates a highly viscous upper boundary layer (the lithosphere), while the rest of the mantle is less viscous. We also implement the decrease of viscosity in the asthenosphere (the layer below the lithosphere) by reducing by a factor of 10 the viscosity  $\mu_n$  when the temperature is above a solidus

5 equation  $T_s = T_{s_0} + \nabla_r T_s(r_a - r)$  with  $r_a$  the surface value of r. The implementation presence of a weak asthenosphere tends to favor plate-like behavior (Tackley, 2000; Richards et al., 2001), and is compatible with laboratory and observational data (King, 2016).

The pseudo-plastic part of the effective viscosity  $\mu_y$  is defined by

$$\mu_y = \frac{\sigma_{yield}}{2\dot{\epsilon}_{\scriptscriptstyle \rm II}},\tag{7}$$

10 where  $\dot{\epsilon}_{II}$  is the second invariant of the strain rate tensor and  $\sigma_{yield} = \sigma_Y + (r_a - r)\nabla_r \sigma_Y$ , with  $\sigma_Y$  and  $\nabla_r \sigma_Y$  the yield stress at the surface and the depth-dependence of the yield stress, respectively.

This composite rheology allows the development of strong plates delimited by narrow weak zones (i.e. plate boundaries), and is currently the best way to generate self-consistently plate-like tectonics at the surface of global mantle convection models (Coltice et al., 2017).

15 The energy conservation equation is the only prognostic equation of the system

$$\frac{DT}{Dt} = \nabla^2 T + \underline{\mathbf{R}_h} \frac{\mathbf{R}_{\mathbf{a}_H}}{\mathbf{R}_{\mathbf{a}_T}}.$$
(8)

[revised manuscript text omitted]
  $\{x_{kn}^f\}_{n\in[1,N]}$  and  $\{x_{kn}^a\}_{n\in[1,N]}$  are computed, such that their average equals  $x_k^f$  and  $x_k^a$ , respectively, and their respective sample covariance matrices approximate  $\mathbf{P}_k^f$  and  $\mathbf{P}_k^a$ . The ensemble of states  $\{x_{kn}^f\}_{n\in[1,N]}$  and  $\{x_{kn}^a\}_{n\in[1,N]}$  are stored in the matrices  $\mathbf{X}_k^f$  and  $\mathbf{X}_k^a$ , where the *n*th column is the state of the *n*th ensemble member  $x_{kn}^f$  and  $x_{kn}^a$ , respectively.

Finally, we introduce the observation operator, which maps a given state vector  $\boldsymbol{x}_{kn}^e$  (*e* being *f* or *a*) to the corresponding data  $\boldsymbol{y}_{kn}^e$ . If the The surface heat flux is approximated by a first order discretization of Fourier's law, then the . The observation operator is linear, then linear, with its velocity part being simply the identity, and can be represented by the matrix **H** such that

$$\forall k \in \{1, 2, ..., K\}, \forall n \in \{1, 2, ..., N\}, \quad \boldsymbol{y}_{kn}^e = \mathbf{H}\boldsymbol{x}_{kn}^e.$$
(12)

10 Table 2 summarizes the dimensions of the vectors and matrices for our problem.

**3 Ensemble Kalman filter with localization and inflation**

The ensemble Kalman filter (Evensen, 1994; Burgers et al., 1998) is a sequential data assimilation algorithm using the same equations as the Kalman Filter for the analysis step, but Monte Carlo methods to forecast the error statistics on the state. We explain here how we adapt the ensemble Kalman filter to our problem and justify the choice of the starting ensemble.

15

5

To implement the EnKF, we used the software environment Parallel Data Assimilation Framework (PDAF, Nerger et al., 2005; Nerger and Hiller, 2013).

**3.1 Initialization: first analysis and generation of the starting ensemble**

As in Bocher et al. (2016), we compute We compute the second order statistics of the background state from a series of 400 decorrelated snapshots of convection simulations . We obtain by following the procedure detailed in Bocher et al. (2016),

20 Sect. 4.1. The model setup is spherically symmetric, so the expected value and covariance of the background temperatures and surface velocities must satisfy

$$\forall (\phi, r), \langle T(\phi, r) \rangle = \langle T(0, r) \rangle, \tag{13}$$

$$\forall (\phi_1, \phi_2, r_1, r_2), \operatorname{Cov}(T(\phi_1, r_1), T(\phi_2, r_2)) = \operatorname{Cov}(T(0, r_1), T(\phi_1 - \phi_2, r_2)),$$
(14)

$$= \operatorname{Cov}(T(0, r_1), T(\phi_2 - \phi_1, r_2)), \tag{15}$$

where  $\langle \cdot \rangle$  stands for the expectation operator and  $Cov(\cdot, \cdot)$  stands for covariance operator. Likewise, we have

$$\forall \phi, \langle u_{\phi}(\phi) \rangle = \langle u_{\phi}(0) \rangle, \tag{16}$$

$$\forall (\phi_1, \phi_2), \operatorname{Cov}(u_{\phi}(\phi_1), u_{\phi}(\phi_2)) = \operatorname{Cov}(u_{\phi}(0), u_{\phi}(\phi_1 - \phi_2)), \tag{17}$$

$$\forall (\phi_1, \phi_2, r_1), \operatorname{Cov}(T(\phi_1, r_1), u_{\phi}(\phi_2)) = \operatorname{Cov}(T(0, r_1), u_{\phi}(\phi_2 - \phi_1)),$$
(18)

$$= -\text{Cov}(T(0, r_1), u_{\phi}(\phi_1 - \phi_2)).$$
(19)

We use these symmetries to compute

5

$$\langle T(0,r_m)\rangle$$
, with  $m \in \{1,...,M\}$  (20)

$$\underbrace{\text{Cov}(T(r_m), T(\phi_{l'}, r_{m'})), \text{ with } m \in \{1, ..., M\}, l' \in \{1, ..., L/2\}, \text{ and } m' \in \{1, ..., M\}}_{\leftarrow}$$
(21)

$$\langle u_{\phi}(0) \rangle,$$
 (22)

10
$$\operatorname{Cov}(u_{\phi}(0), u_{\phi}(\phi_l)), \text{ with } l \in \{1, \dots, L/2\}$$
 (23)

$$Cov(u_{\phi}(0), T(\phi_l, r_m)), \text{ with } l \in \{1, ..., L/2\}, \text{ and } m \in \{1, ..., M\},$$
(24)

and build with these values the first forecast state of average expected value  $x_1^f$  and associated covariance matrix  $\mathbf{P}_1^f$ . The background For the model used in this study (see Table 1), the covariance matrix  $\mathbf{P}_1^f$  has  $(LM + L)^2 = 18,816^2 = 354,041,856$  components. By using the symmetries in the system, we are able to reduce the number of independant components in the

15 covariance matrix to  $L/2(M+1)^2 = 3,557,400$ .  $\mathbf{P}_1^f$  is eigendecomposed and rank reduced into  $\mathbf{P}_{1r}^f = \mathbf{V}\mathbf{A}\mathbf{V}^T$ , with  $\mathbf{A}$  $\mathbf{P}_{1r}^f = \mathbf{V}\mathbf{A}\mathbf{V}^T$ , with  $\mathbf{A}$  a diagonal matrix containing the 1928  $n_r = 1928$  largest eigenvalues of  $\mathbf{P}_1^f$  (which accounts for 99.98% of its cumulative variance) and  $\mathbf{V}$  a matrix of the corresponding the corresponding matrix of eigenvectors.

The We assimilate the first set of observations  $y_1^o$  is assimilated to obtain using the classical Best Linear Unbiased Estimator equations (see Ghil and Malanotte-Rizzoli (1991) for example). When the forecast covariance matrix is eigendecomposed and rank reduced, these equations can take the form

$$\boldsymbol{x}_{1}^{a} = \boldsymbol{x}_{1}^{f} + \mathbf{V}\mathbf{A}\mathbf{V}^{T}\mathbf{H}^{T}\mathbf{R}^{-1}(\boldsymbol{y}_{1}^{o} - \mathbf{H}\boldsymbol{x}_{1}^{f}),$$
(25)

$$\mathbf{P}_1^a = \mathbf{V} \mathbf{A} \mathbf{V}^T,\tag{26}$$

with

20

$$\mathbf{A} = \left[\mathbf{\Lambda}^{-1} + \mathbf{V}^T \mathbf{H}^T \mathbf{R}^{-1} \mathbf{H} \mathbf{V}\right]^{-1}.$$
(27)

25 We After the first analysis, we generate an ensemble of N initial states using from the first analyzed state average  $x_1^a$  and associated covariance matrix  $\mathbf{P}_1^a$ . To do so, we follow the second order exact sampling method (Hoteit, 2001; Pham, 2001). First, A is eigendecomposed

$$\mathbf{A} = \mathbf{V}^a \mathbf{\Lambda}^a \mathbf{V}^{aT}.$$
(28)

The ensemble members are then computed following

$$\mathbf{X}_{1}^{a} = \begin{pmatrix} | & | \\ \boldsymbol{x}_{11}^{a} & \dots & \boldsymbol{x}_{1N}^{a} \\ | & | \end{pmatrix} = \begin{pmatrix} | & | \\ \boldsymbol{x}_{1}^{a} & \dots & \boldsymbol{x}_{1}^{a} \\ | & | \end{pmatrix} + \sqrt{N-1} \mathbf{V} \mathbf{V}^{a} \mathbf{\Lambda}^{a1/2} \begin{pmatrix} \mathbf{\Omega}_{N \times (N-1)}^{T} \\ \mathbf{0}_{(n_{r}-N) \times N} \end{pmatrix},$$
(29)

where  $\Omega \Omega_{N \times (N=1)}$  is a random matrix whose columns are vectors forming an orthonormal basis and each of them is orthogonal to  $\mathbf{1} = [1, ..., 1]^T$ .  $\Omega \mathbf{1}_N$ , the column vector of dimension N full of 1,  $\mathbf{1}_N = [1, ..., 1]^T$ .  $\mathbf{0}_{(n_r = N) \times N}$  is a  $(n_r - N) \times N$  matrix full of 0.  $\Omega_{N \times (N=1)}$  is generated through the algorithm described in the appendix of Nerger et al. (2012). The matrix

5 matrix full of 0.  $\Omega_{N \times (N-1)}$  is generated through the algorithm described in the appendix of Nerger et al. (2012). The matrix  $\Omega_{N \times (N-1)}$  is designed so that the sample mean of the starting ensemble is equal to  $x_1^a$  and its sample covariance matrix is equal to matrix  $\mathbf{P}_1^a$  reduced to its N largest eigenvalues.

This method of generating the starting ensemble takes advantage of the extensive knowledge we have on the background statistics of the model. Several other methods have been tested to generate a starting ensemble, such as starting with random decorrelated snapshots of mantle convection obtained from a very long runsimulations, second order exact sampling from  $x_1^f$ and  $\mathbf{P}_1^f$ , and several assimilations of the first observations  $y_1^o$ . None of these solutions were as efficient for our problem as the technique used here. These alternative solutions resulted in reconstructions with larger initial errors and slower error decrease throughout the assimilation window, if any.

**3.2 Forecast**

15 Between time steps k-1 and k, the forward numerical code STAGYY computes independently the evolution of each of the analyzed states  $\{x_{k-1,n}^a\}_{n \in [1,N]}$  to produce a forecast ensemble  $\{x_{k,n}^f\}_{n \in [1,N]}$ .

The forecast state is the average of the ensemble

$$\boldsymbol{x}_{k}^{f} = \frac{1}{N} \mathbf{X}_{k}^{f} \mathbf{1}_{\underbrace{N}}.$$
(30)

and the The forecast error covariance matrix is given by the sample covariance matrix of the ensemble of forecast states

20
$$\mathbf{P}_{k}^{f} = \frac{1}{N-1} \mathbf{X}_{k}^{f} \left( \mathbf{I}_{\underline{N}} - \frac{1}{N} \mathbf{1}_{\underline{N}} \mathbf{1}_{\underline{N}}^{T} \right) \left( \mathbf{I}_{\underline{N}} - \frac{1}{N} \mathbf{1}_{\underline{N}} \mathbf{1}_{\underline{N}}^{T} \right)^{T} \mathbf{X}_{k}^{fT}$$
(31)

where  $I_N$  is the identity matrix of dimension  $N \times N$ . After several assimilation cycles, the finite size of the ensemble induces the underestimation of the error variance (van Leeuwen, 1999), and can lead to filter divergence. We observed this behavior in our case, and to stabilize the filter we apply covariance inflation, as suggested in Anderson and Anderson (1999) and Hamill et al. (2001).

25 We correct the forecast ensemble variance with an inflation factor  $\gamma$  according to

$$\mathbf{X}_{k}^{f} \leftarrow \frac{1}{N} \mathbf{X}_{k}^{f} \mathbf{1}_{\underline{N}} \mathbf{1}_{\underline{N}}^{T} + \left[ \mathbf{X}_{k}^{f} \left( \mathbf{I}_{\underline{N}} - \frac{1}{N} \mathbf{1}_{\underline{N}} \mathbf{1}_{\underline{N}}^{T} \right) \right] \sqrt{\gamma},$$
(32)

where  $\leftarrow$  means that we replace the matrix on the left-hand side by the term on the right-hand side.  $\gamma$  is computed following the same principles as in the suboptimal Kalman Filter developed in Bocher et al. (2016), i.e. by comparing the error on

observations and the standard deviation of the innovation  $d_k$  defined as

$$\boldsymbol{d}_{k} = \boldsymbol{y}_{k}^{o} - \frac{1}{N} \mathbf{H} \mathbf{X}_{k}^{f} \mathbf{1}_{\underline{N}}.$$
(33)

The inflation factor is

$$\gamma = \frac{V^d - V^o}{V^f},\tag{34}$$

5 with

$$V^{d} = \operatorname{Tr}\left(\boldsymbol{d}_{k}\boldsymbol{d}_{k}^{T}\right),\tag{35}$$

$$V^o = \operatorname{Tr}(\mathbf{R}_k),\tag{36}$$

$$V^{f} = \operatorname{Tr}\left[\mathbf{H}\mathbf{X}_{k}^{f}\left(\mathbf{I}_{\underline{N}} - \frac{1}{N}\mathbf{1}_{\underline{N}}\mathbf{1}_{\underline{N}}^{T}\right)\left(\mathbf{I}_{\underline{N}} - \frac{1}{N}\mathbf{1}_{\underline{N}}\mathbf{1}_{\underline{
[revised manuscript text omitted]
})}} \qquad \text{and} \qquad \epsilon_{u_{\phi}}^{e}(k) = \sqrt{\frac{1}{2} \left(\frac{1}{2} \sum_{l=1}^{M} \frac{1}{2} \sum_{l=1}^{M} \frac{1}{2} \left(\frac{1}{2} \sum_{l=1}^{M} \frac{1}{2} \left(\frac{1}{2} \sum_{l=1}^{M} \frac{1}{2} \left(\frac{1}{2} \sum_{l=1}^{M} \frac{1}{2} \sum_{l=1}^{M} \frac{1}{2} \left(\frac{1}{2} \sum_{l=1}^{M} \frac{1}{2} \left(\frac{1}{2} \sum_{l=1}^{M} \frac{1}{2} \left(\frac{1}{2} \sum_{l=1}^{M} \frac{1}{2} \sum_{l=1}^{M} \frac{1}{2} \left(\frac{1}{2} \sum_{l=1}^{M} \frac{1}{2} \left(\frac{1}{2} \sum_{l=1}^{M} \frac{1}{2} \sum_{l=1}^{M} \frac{1}{2} \sum_{l=1}^{M} \frac{1}{2} \left(\frac{1}{2} \sum_{l=1}^{M} \frac{1}{2} \sum_{l=1}^{M} \frac{1}{2} \sum_{l=1}^{M} \frac{1}{2} \sum_{l=1}^{M} \frac{1}{2} \left(\frac{1}{2} \sum_{l=1}^{M} \frac{1}{2} \sum_{$$

20

with  $\mathcal{V}(\phi_l, r_m)$  the volume of the grid cell at longitude  $\phi_l$  and radius  $r_m$ , and  $\overline{T}_k^e(\phi_l, r_m)$  the average temperature and  $\overline{u}_{\phi k}^e(\phi_l)$ . the average horizontal velocity of the estimated ensemble (either forecast or analysis) at longitude  $\phi_l$  and radius  $r_m$  and  $r_a$ , and where the superscript t still refers to the true state.

We test the EnKF on one evolution, with sizes of the ensemble N = 96, 288 and 768 and for each combination of the 25 following values of the data assimilation parameters values: vertical correlation length  $\ell_v = 0.3$ , 0.5, 0.7 and 1 and horizontal correlation angle  $\ell_h = \pi/10$ ,  $\pi/8$ ,  $\pi/6$ ,  $\pi/4$  and  $\pi/2$ . We show in Fig. 1, for each ensemble size, the maximum and minimum values of errors on temperature (Fig. 1(a-c)) and on surface horizontal velocity (Fig. 1(d-e)), obtained for all these parameters, as a function of time. We also represent the background error on temperature  $\epsilon_T^b(k)$  and on surface horizontal velocity  $\epsilon_{u+}^b(k)$

$$\epsilon_{T}^{b}(k) = \sqrt{\frac{\sum_{l=1}^{L} \sum_{m=1}^{M} \left(T^{b}(r_{m}) - T_{k}^{t}(\phi_{l}, r_{m})\right)^{2} \mathcal{V}(\phi_{l}, r_{m})}{\sum_{l=1}^{L} \sum_{m=1}^{M} \mathcal{V}(\phi_{l}, r_{m})}} \quad \text{and} \quad \epsilon_{u_{\phi}}^{b}(k) = \sqrt{\frac{\sum_{l=1}^{L} \left(u_{\phi}^{b}(r_{a}) - u_{\phi k}^{t}(\phi_{l}, r_{a})\right)^{2} \mathcal{V}(\phi_{l}, r_{a})}{\sum_{l=1}^{L} \mathcal{V}(\phi_{l}, r_{a})}} \quad (42)$$

where  $T^b$  and  $u^b_{\phi}$  are 1D profiles corresponding to the average temperature and horizontal velocity, respectively, computed from a long run.

5 To determine the best assimilation, we compute We choose the average error after analysis on temperature after analysis

$$\bar{\epsilon}_T^a = \frac{1}{K} \sum_{k=1}^K \epsilon_T^a(k) \underline{.}$$
(43)

as the global measure for the quality of the assimilation. For each ensemble size, the error evolution of the best assimilation (in the sense of minimum  $\overline{\epsilon}^a_T \overline{\epsilon}^a_T$ ) is also shown in Fig. 1.

For The error evolutions for temperature and surface horizontal velocity follow the analysis-forecast sequence: at each analysis time (every 10 Myrs), the error decreases abruptly, and during the forecast phases, the error increases.

10

For the surface horizontal velocity (Fig. 1(d-f)), the error evolutions are very similar regardless of the data assimilation parameters: the error decreases drastically during the analysis to a value of 25 to 50, while the amplitude of the error growth during the forecast phase evolves from around 200 for the first forecasts to around 100 at the end of the assimilation.

- On the contrary, the evolution of the error on the temperature depends on the parameters of the assimilation. Fig. 1(a-c) shows that, for any size of the ensemble, it is possible to find a set of parameters leading to a stabilization drastic reduction of the global error on the temperature field after a few analyses. The time after which the solution is stabilized This first phase, when errors decrease quickly, lasts approximately 70 Myr, which corresponds to the transit time of the physical model (70 Myr)dynamic system. After this phase, the error on temperature slowly increases with time, while remaining well below the errors measured for the first analyses. We can see that for N = 288 and N = 768, any combination of vertical and horizontal correlation
- 20 lengths leads to errors lower than the first analysis. Although the error is decreasing through time for any combination of data assimilation parameters. However, the difference between the maximum and the minimum errors obtained is greater than 1%0.01, which is large given that the first analysis error is already below 8%. considering the background error is only around 0.1. The best error evolutions for N = 288 and N = 768 are very similar, with a minimum error of 4.07% and 3.87% 0.0318 and 0.0302 after 90 Myr, and an average global error after analysis of 5.01% and 4.85%0.0391 and 0.0378, respectively.
- During the assimilation of a dataset, most of the computational time is dedicated to the forecast step, so the data assimilation with 768 members is 2.7 times longer more expensive (computationally speaking) than the assimilation with 288 members, on the account of the embarrassingly parallel nature of the forecast phase. Since we obtain very similar results for N = 288 and N = 768, we favor the assimilation with 288 members.

We compute the error on the estimated temperature from by comparing it to the true temperature field. However, in a realistic case, the true temperature is not known, and the evaluation of the data assimilation algorithm is based on the study of

---

## Editor Decision (ED1)

Chère Collègue,

I have now received the evaluation of two referees on the new version of your paper. The referees are the same as those of the first version. In particular, referee 1, who has now let his name known, is Jeffrey Anderson.

Both referees suggest acceptance of your paper as it is. Jeffrey Anderson adds comments on the question of localization. I follow their opinion, and accept the paper. I however suggest, as editor, two corrections.

1. It seems that the parameters that are introduced in subsection 2.1 (*Mantle convection model*) are not all properly defined. For instance, I understand that $a^E$ (p. 5, l. 28) is the same as $a$ (p. 4, l. 13), *i.e.* $r_a - r_b$ in Table 1. Please check carefully that all parameters are properly defined, and their numerical values specified.

2. You write (p. 7, ll. 18-19) *We choose to include in the state the whole temperature field, but also add the surface velocities, to form an augmented state vector, ....* I understand that this means that the analyzed surface velocities are used only as diagnostic quantities, and have no impact on the sequel. But it also means that the analyzed temperature and surface velocities are not linked by Eqs (1-2). If so, say it clearly. If not, explain.

I thank you for having submitted your paper to *Nonlinear Processes in Geophysics*, and look forward to receiving your final version,

Cordialement,

Olivier Talagrand
Editor
*Nonlinear Processes in Geophysics*